

# Satellite-based estimation of contrail cirrus cloud radiative forcing derived through a Rapid Contrail-RF Estimation Approach

Ermioni Dimitropoulou ●[1], Pierre de Buyl ●[1], and Nicolas Clerbaux ●[1]

[1]Royal Meteorological Institute of Belgium, 1180 Brussels, Belgium

**Correspondence:** Ermioni Dimitropoulou (ermioni.dimitropoulou@meteo.be)

**Abstract.** Geostationary satellite observations were used to estimate the radiative forcing of contrail cirrus clouds through a Rapid Contrail-RF Estimation Approach. Meteosat Second Generation/ Spinning Enhanced Visible and InfraRed Imager (MSG/SEVIRI) observations were utilized to visually identify days with contrails. For six selected days, ice clouds were characterized using the Optimal Cloud Analysis (OCA) product from MSG/SEVIRI data provided by the European Organisation for the Exploitation of Meteorological Satellites (EUMETSAT). Look-Up Tables (LUTs) were constructed using the libRadtran radiative transfer (RT) model to quantify the radiative effect of ice clouds in the shortwave (SW) and longwave (LW) spectral regions. The retrieved cloud properties were combined with the LUTs to generate radiative forcing maps for natural and contrail cirrus clouds. A separation scheme isolated the radiative forcing of contrails. The resulted dataset provides a quantification of the SW, LW and net radiative forcing at the top of the atmosphere (TOA) due to contrails. Over the full diurnal cycle, contrails cause a cooling effect during the daytime and warming at night. The Rapid Contrail-RF Estimation Approach's validity was assessed through correlative exercises focusing on uncertainties in the use of LUTs, a single ice cloud parameterization, and a calculated cloud top height (CTH), supplemented by comparisons with polar-orbiting satellite observations from the Clouds and the Earth's Radiant Energy System (CERES) instruments. Overall, these correlative comparisons indicate that the proposed approach provides accurate data on contrails radiative forcing estimation, with an accuracy on the order of approximately 15 %.

## 1   Introduction

Understanding the role of clouds in the Earth's radiation budget is crucial for mitigating climate change Wielicki et al. (1995). The latest IPCC report highlights that clouds and aerosols still represent the largest source of uncertainty to estimates and interpretations of the Earth's energy budget, particularly when these clouds are caused by anthropogenic activities such as aviation (Lee et al., 2023). Aviation contributes approximately 5% to the anthropogenic climate forcing with the emission of carbon dioxide ($CO_2$) and non-$CO_2$ pollutants to be the two main contributors (Lee et al., 2009, 2021). Since $CO_2$ emissions are known to be the primary reason of observed global warming (Letcher, 2020), many studies and reports were focusing primarily on the quantification of aviation's contribution to the global atmospheric $CO_2$ concentrations (Olsthoorn, 2001; Pejovic et al., 2008; Ji-Cheng and Yu-Qing, 2012; Mayor and Tol, 2010; Howitt et al., 2011). However, the historical lack of



focus on non-$CO_2$ emissions is not due to their insignificance for climate, but rather because they are not yet fully understood and remain associated with considerable uncertainty (Lee et al., 2021).

The non-$CO_2$ aviation pollutants include emissions of nitrogen oxides ($NO_x$ = NO + $NO_2$), water vapor ($H_2O$), soot, sulfur oxides ($SO_x$) and the formation of contrail cirrus clouds (Lee et al., 2021). Among these, contrails most likely have the largest impact on the TOA radiation budget Burkhardt and Kärcher (2011); Brasseur et al. (2016). These aviation-induced clouds are

formed behind aircraft engines when the mixture of exhaust gases and ambient air arrives in saturation with respect to liquid water during the plume expansion (isobaric) process (Gierens et al., 2008). Young contrails persist and spread into larger clouds when they are formed in ice supersaturated regions (ISSRs) (Schumann et al., 2017; Unterstrasser, 2020). The properties of these young contrails, together with their geometric depth and total ice crystal number will affect the later properties of the persistent contrail cirrus clouds (Unterstrasser, 2016). Only 10-15 % of contrails persist as contrail cirrus, with an average

lifetime of approximately 4 hours (Gierens and Vázquez-Navarro, 2018). These persistent contrails are the only ones relevant for changes in the Earth's radiation budget.

The impact of contrails on the TOA radiation budget is often quantified using the radiative forcing (RF) metric (Chen et al., 2000), which is defined as the change in TOA energy budget due to a constituent in absence of (almost) any other change. In the solar wavelength range (i.e., SW), contrails have a cooling effect by reflecting incident radiation while in the thermal-infrared

wavelength range (i.e, LW), a warming effect by absorbing and emitting radiation. By summing both radiative forcings, the net radiative effect of cirrus clouds can be calculated.

The estimation of contrails' RF and/or effective RF (ERF) both globally and regionally over extensive time periods is crucial for understanding aviation's contribution to climate change. On a global scale, either general circulation models of the atmosphere, such as the European Centre/Hamburg general circulation model version 4 (ECHAM4) and the Community

Atmosphere Model (CAM5), reanalyses data in combination with radiative transfer modelling, or a combination of a model and observations are used to estimate the global yearly mean contrails' net RF and/or ERF (Rädel and Shine, 2008; Lee et al., 2021; Gettelman et al., 2021; Bock and Burkhardt, 2016; Chen and Gettelman, 2013; Bier and Burkhardt, 2022; Teoh et al., 2023). These studies have reported that the presence of contrails has a yearly positive global net radiative impact, which varies significantly between studies, ranging from 6 $mW/m^2$ (Rädel and Shine, 2008) to 62.1 $mW/m^2$ (Teoh et al., 2023).

On smaller spatial and temporal scales, whether the net effect of contrails is dominated by cooling (negative RF) or warming (positive RF) is highly dependent on the properties of the contrail, as well as the radiative properties of the environment (Wang et al., 2024; Wolf et al., 2023). To address this, the behavior of contrails over different regions, times of day, and seasons should be investigated. To cover the lifetime of persistent contrails and study their diurnal variation, the use of low earth orbit (LEO) satellites provides observations of high spatial resolution but limited temporal coverage (maximum two overpasses per day

over a region) (Spangenberg et al., 2013; Mannstein et al., 1999; Duda et al., 2019). Geostationary satellites, on the other hand, provide high temporal resolution with satellite images available every 15 minutes, as in the case of MSG/SEVIRI observations, while offering high spatial resolution for the portion of the globe they observe. Thus, some studies use polar-orbiting satellite observations together with geostationary satellite observations, while others rely only on geostationary satellite observations





(Haywood et al., 2009; Wang et al., 2024; Dekoutsidis et al., 2023; Duda et al., 2004; Mannstein and Schumann, 2005; Graf
et al., 2012; Schumann and Graf, 2013; Wang et al., 2023; Meijer et al., 2022).

Once the contrails are detected, radiative transfer models can be used to quantify their radiative effect on the TOA. Although
it is not the only way to perform such a task (Haywood et al., 2009), radiative transfer models are a useful tool to quantify
the radiative impact of clouds. In the study of Wang et al. (2024), the quantification of the net effect of contrails is conducted
by performing full radiative transfer calculations with the inputs being the measured cloud and environmental properties for
each satellite pixel over Western Europe for two days. It has been proposed by Wolf et al. (2023) that once the cloud and
environmental properties are well-described, a large number of RT simulations can be performed to construct cirrus clouds,
and contrail cirrus RF LUTs instead of performing full radiative transfer calculations.

The use of RF LUTs combined with geostationary satellite observations offers many advantages but also presents some
drawbacks. By performing a large number of radiative transfer simulations once, while varying relevant parameters, such as
the solar zenith angle, the surface albedo, the ice cloud's optical thickness, etc., we can construct multi-dimensional LUTs,
which describe the behavior of the solar and thermal/infra-red RF ($RF_{sol}$ and $RF_{tir}$, respectively) as a function of these
parameters. Then, these LUTs can be merged with geostationary satellite observations to generate contrails' RF maps. Large
datasets can be processed relatively quickly, enabling the analysis of full years of geostationary data and the study of daily
and seasonal patterns of contrails' RF. Furthermore, once the LUTs are constructed, they are independent of the satellite
instrument's characteristics, allowing them to be merged with different satellite observations. However, the primary concern
with using LUTs is how accurately they represent the real atmospheric conditions for each pixel. Often, the LUTs are built
using standard atmospheric profiles or specific ice cloud properties, which may not capture the variability of real atmospheric
conditions.

In this study, a new satellite-based contrails RF maps approach is presented. We refer to this new approach as Rapid Contrail-
RF Estimation Approach. The Rapid Contrail-RF Estimation Approach combines geostationary satellite observations, a cloud
properties retrieval algorithm, radiative transfer modeling and a separation scheme between natural and contrail cirrus clouds.
First, MSG/SEVIRI observations were employed to visually identify days with the presence of contrails. For the selected days,
the detection and characterization of ice clouds, including those overlapping with lower-layer clouds, were performed using the
OCA product (EUMETSAT, 2019b) derived from MSG/SEVIRI data. Second, LUTs were constructed using the libRadtran RT
model (Emde et al., 2016) to quantify the radiative effects of thin to semi-transparent ice clouds in both the SW (reflected solar
radiation) and LW (thermal radiation) spectral regions. Third, the retrieved cloud properties were combined with the LUTs to
generate radiative forcing maps of both natural and contrail cirrus clouds, with a 15-minute temporal resolution on a regular
grid of spatial resolution equal to $0.04^o$. Finally, a separation scheme was applied to distinguish between natural and contrail
cirrus clouds, isolating the radiative forcing specific to contrails. The dataset covers six different days within the 2023-2024
period on a geographic area expanding from $30^oW$ to $15^oE$ longitude and $25^oN$ to $55^oN$ of latitude. We used the generated
RF maps of contrails to investigate their behavior with respect to the TOA radiation budget. The uncertainty of the Rapid
Contrail-RF Estimation Approach has been assessed via three different validation exercises by evaluating (1) the choice of
using a single atmospheric vertical profile in the RT simulations, (2) the choice of a single ice cloud parameterization scheme,





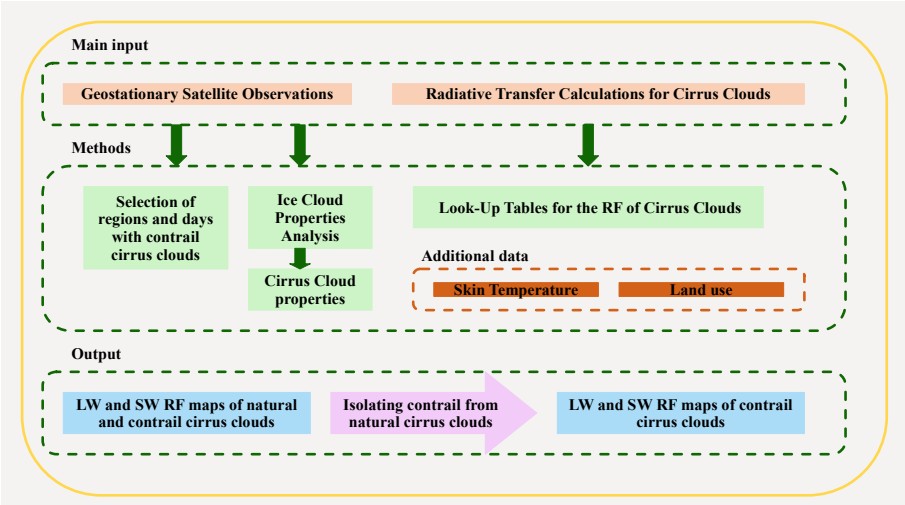

**Figure 1.** Methodological flowchart describing the Rapid Contrail-RF Estimation Approach.

and (3) the impact of using CTH values estimated by a single atmospheric vertical profile on the RF estimations. Additionally,
an end-to-end validation is performed by comparing the flux maps for contrails and polar-orbiting satellite observations from
the CERES instruments.

The paper is organized into five sections: Section 2 presents the Rapid Contrail-RF Estimation Approach for generating RF
maps for contrail cirrus clouds, followed by the necessary data. In Section 3 a description of the methodology for merging the
different datasets is provided. The main results of the study, along with four different validation exercises to assess the validity
and estimate the uncertainty of the Rapid Contrail-RF Estimation Approach, are discussed in Section 4. Finally, conclusions
and future perspectives are provided in Section 6.

## 2   Data for the Rapid Contrail-RF Estimation Approach

In this study, generating RF maps for contrails above the geographic area of interest follows the Rapid Contrail-RF Estimation
Approach composed of the initial three steps: (1) detection (Subsection 2.1), (2) characterization (Subsection 2.2), and (3)
estimation of the RF (Subsection 2.3) for cirrus clouds.

Additionally, different datasets have been employed for two main purposes: (1) to accurately describe the conditions and
characteristics of the geographic area of interest (see Subsections 2.4 and 2.5) and (2) to validate the RF maps of contrails (see
Subsections 2.4 and 2.6). A flowchart describing the Rapid Contrail-RF Estimation Approach is shown in Figure 1.



## 2.1 Geostationary Satellite Observations

The detection and characterization of ice clouds was carried out using data from the SEVIRI on board MSG-3 (Meteosat-10) satellite, which is operated by EUMETSAT. MSG-3 is located at $0^o$ longitude in geostationary orbit, approximately 36.000 kilometres above the Earth's surface.

SEVIRI provides spectral information across 11 spectral channels covering the visible and infrared spectral regions (Schmetz et al., 2002). These observations are captured every 15 minutes. The spatial resolution of SEVIRI is approximately 3x3 $km^2$

at the sub-satellite point (SSP), which can become larger due to the satellite's viewing angle. There is an additional channel, the broad-band High-Resolution Visible (HRV), which has a higher spatial resolution compared to the other spectral channels (1x1 $km^2$), which is not used in cloud retrieval algorithms.

Our primary objective is to investigate the behavior of contrails above different surface types, seasons, and times of the day. These clouds can be visually detected using the Dust/RGB (Red, Green, Blue) composite, which combines data from

the MSG/SEVIRI IR8.7, IR10.8 and IR12.0 channels. The Dust product is initially designed to monitor the evolution of dust storms during day and night. Additionally, it is also useful for the detection of thin cirrus clouds. In this product, the contrails appear as long bluish and reddish lines and are visually distinguished by other cloud types. Previous studies such as Wang et al. (2023); Schmetz et al. (2002); DEKOUTSIDIS and FEIDAS (2019), utilized successfully the Dust/RGB composite to detect contrail cirrus clouds. In this study, using this composite, with the aid of the Satpy Python library (Hoese, 2019) and

the EUMETSAT RGB recipes (https://eumetrain.org/manualguides/rgb-recipes), we identified specific days during which we could visually detect geographic regions above Europe and parts of the North Atlantic Ocean where persistent contrails are present.

In summary, the study area extends from $30^o$W to $15^o$E of longitude and $25^o$N to $55^o$N of latitude covering data from six different days: 30-01-2023; 13-06-2023; 25-09-2023; 30-01-2024; 17-02-2024; and 28-05-2024. The geostationary grid has

been re-projected onto a regular grid with a spatial resolution of $0.04^o$.

## 2.2 Cloud Analysis Product

In the present study, the OCA product is used for the physical characterization of the detected ice clouds.

The OCA algorithm uses the Optimal Estimation (OE) method along with spectral measurements simultaneously to retrieve the cloud state parameters (Rodgers, 2000). The primary concept behind the OE method is to determine the state vector that

closely matches the real measurement vector and the simulated measurement vector. A priori information of the atmospheric observations is used together with the forward model to simulate the observations that would be linked to the state vectors. This is done because the problem is ill-posed, which means that more than one state vector can be associated to the observations that we have. Consequently, by using a priori constrains, we select the most likely state which matches the measurements (Rodgers, 2000).

Within OCA, the state vector of the retrieval, which describes the state of the atmosphere, contains six variables: the cloud optical thickness (COT), cloud-top particle effective radius (CER), cloud-top pressure (CTP), cloud fraction, cloud phase





and surface skin temperature (SST) (Mecikalski et al., 2011). The measurement vector, which is used within OCA, includes the following SEVIRI channels: the reflectance percentage (0 - 1) in the 0.6, 0.8, and 1.6 $\mu$m channels and the brightness temperature in the 6.2, 7.3, 8.7, 10.8, 12.0, and 13.4 $\mu$m channels. For OCA, the forward model is a radiative transfer model

(RTM) and more precisely, LUTs obtained by a fast RTM (FASTRTM). The RTM in the OE method also depends on Numerical Weather Prediction (NWP) data: humidity, temperature and gaseous constituents' profiles, surface albedo and emissivity at the IR channel wavelengths, and other radiative properties (EUMETSAT, 2016).

The OCA product is available at a 15-minute frequency and at the full earth scanning area in GRIB format (EUMETSAT, 2019a). It contains single-layer as well as multi-layer cloud situations and the product is structured in layers numbered in a

top-down notation: the first layer (named Layer-One) is the highest layer (closest to the top of the atmosphere) and the second layer (named Layer-Two) is the lower layer, which only exists when the pixel has multi-layer cloud conditions (Watts et al., 2011). It should be noted that for the multi-layer cloud scenes, the upper layer is assumed to be an ice cloud and the lower layer a liquid cloud.

For the present study, we will focus only on pixels characterized as ice or multi-layered clouds (cloud phase) and will use

both COT and CTP for the two layers and CER only for the ice cloud. An additional parameter is CTH, which is not included in the OCA product. As we are using a US Standard atmosphere for the construction of the LUTs (see Section 2.3), the same profile has been used to linearly interpolate CTP in the pressure vertical grid, and consequently the altitude vertical grid, of the US Standard atmosphere (Anderson et al., 1986). The uncertainty related to the use of a calculated CTH is presented in Section 5.0.3.

**2.3    Radiative Transfer Calculations**

LUTs of thin to semi-transparent high-altitude ice clouds are constructed to quantify the contrail cirrus clouds' RF. The simulations are conducted using the libRadtran software (Emde et al., 2016).

The libRadtran RT library (version 2.0.5) is used to construct LUTs of TOA irradiances for the SW (reflected solar radiation) and LW (emitted thermal radiation) spectral regions, separately. The RT simulations were performed with the one-dimensional

(1-D) DIScrete ORdinate Radiative Transfer (DISORT) solver (Stamnes et al., 2000), which is included in the libRadtran software package. Using a 1-D solver means that the ice clouds are assumed to be horizontally uniform.

The spectra for both the SW and LW wavelength regions are simulated under three different scene scenarios:

1. Ice cloud above ocean surface,

2. Ice cloud above land surfaces, and

3. Ice cloud above a water cloud.

and, the following input parameters are chosen:

– 16 streams to solve the RT equation, which provides accurate results and limits the computational time.



  – a US-standard atmosphere as the main atmospheric profile. The accuracy of such simplification is assessed in Section 5.0.1.

– the ice crystal shape to be moderately rough aggregates of eight-element columns based on the ice cloud parameterization of Yang et al. (2013). The uncertainty related to the use of a single ice crystal shape scheme is discussed in Section 5.0.2.

  – the ice water content and effective particle radius to be translated to optical properties based on the ice cloud parameterization of Yang et al. (2013).

  – the built-in International Geosphere Biosphere Programme (IGBP) library which is a collection of spectral albedos of
different surface types.

  – the liquid water content and effective radius of the water cloud to be translated to optical properties based on the parameterization of Hu and Stamnes (1993).

In the SW spectral region, the spectra are simulated from 250 nm up to 5000 nm, while in the LW wavelength region, from 2500 nm up to 98000 nm.

For the LW spectral region and an ice cloud above land and ocean surfaces, the molecular absorption is considered by using the fine resolution REPTRAN parameterization from Gasteiger et al. (2014). For an ice cloud above a water cloud, the same molecular absorption is used but at medium resolution.

Six examples of libRadtran input files (i.e., one in the SW and one in the LW for the three scene scenarios) are provided on the Zenodo platform.

The initial output files of the RT simulations in the SW and LW spectral regions contain the wavelength, the output altitude, the direct beam irradiance with respect to the horizontal plane, the diffuse down irradiance and the diffuse up irradiance. These variables are integrated over the total simulation wavelength range separately in the SW and LW wavelength regions to estimate the TOA downward and upward solar/thermal-infrared irradiances ($F_{down}$ and $F_{up}$, respectively).

$RF_{sol}$ and $RF_{tir}$ of ice clouds are defined as the difference in fluxes between the ice cloud ($F_{ic}$) and ice cloud-free ($F_{icf}$)
atmosphere at the TOA:

$$RF = F_{ic} - F_{icf} = [F_{down} - F_{up}]_{ic} - [F_{down} - F_{up}]_{icf} \tag{1}$$

Then, the net RF is a summation between $RF_{sol}$ and $RF_{tir}$ (Wolf et al., 2023). It should be noted that negative $RF$ values indicate cooling, while positive ones correspond to warming.

In the SW wavelength range, the LUTs store the TOA $F_{down}$, $F_{up}$, and $RF_{sol}$ as a function of the following parameters:

– the Solar Zenith Angle (SZA), COT and CER for an ice cloud above ocean surface,

  – the SZA, COT, CER and underlying surface type as defined by the IGBP for an ice cloud above land surface, and

  – the SZA, COT, CER and water Cloud Optical Thickness (wCOT) for an ice above a water cloud.




while keeping constant the following:

– the CTH equal to 10km for the three scene scenarios,

– the ocean Bidirectional Reflectance Distribution Function (BRDF) determined by a wind speed equal to 5 m/s for an ice cloud above ocean surface from Cox and Munk (1954a, b),

– the water Cloud Top Height (wCTH) equal to 3km for an ice cloud above a water cloud.

The LUTs in the LW store the TOA $F_{down}$, $F_{up}$, and $RF_{tir}$ by varying the following parameters:

– the SST, CTH, COT, and CER for an ice cloud above ocean surface,

– the underlying LST, CTH, COT, CER and IGBP surface type for an ice cloud above land surface,

– the wCOT, wCTH, CTH, COT, and CER for an ice cloud above a water cloud.

To save computational time, RT simulations have been performed for all IGBP surface types in the SW spectrum but not in the LW spectrum. In the SW spectrum, there are 17 different LUTs, each corresponding to simulations for the 17 different IGBP surface types. The only IGBP surface type not included in these simulations is type 17 (ocean water). For this type, the LUTs correspond to the scenario of an ice cloud above an ocean surface. In the LW, RT simulations were conducted for 8 specific IGBP surface types (i.e., evergreen needle forest, closed shrubs, open shrubs, woody savanna, grassland, urban, antarctic snow, and desert). The other IGBP surface types were mapped to the existing ones based on their similar or closely related emissivity responses.

The LUTs are all stored in a single NetCDF file and are available on the Zenodo platform.

The reader should note that the RF values stored in the different LUTs depend on at least three parameters (e.g., SW wavelength range and the scene scenario of ice cloud above ocean) and up to five parameters (e.g., LW wavelength range and the scene scenario of ice cloud above a water cloud). It is complex to illustrate every single dependency of RF with respect to each input parameter. Therefore, we have chosen to demonstrate some of these dependencies in Figures 2 and 3.

Figure 2 shows an example of the variation of $RF_{sol}$ for an ice cloud above ocean as a function of COT, SZA (different plots), and CER (different lines in every plot). In the SW wavelength range, RF is negative, corresponding to a cooling effect of the climate due to cloud reflectivity. As observed, $RF_{sol}$ increases in magnitude as the ice cloud becomes thicker (larger COT values). Additionally, the size of the ice crystals (i.e., CER) becomes significant for thick clouds. Finally, as we can see, the SZA also plays an important role in the simulations, resulting in larger absolute $RF_{sol}$ values for a SZA value equal to $0^o$.

Similar to Figure 2, Figure 3 shows the variation of $RF_{tir}$ for an ice cloud above the ocean as a function of COT, SST (different plots), and CER (different lines in every plot). Here, we observe that $RF_{tir}$ changes rapidly for small values of COT, specifically in the range from 0 to 10. However, $RF_{tir}$ saturates when the cloud reaches a certain thickness (approximately a COT equal to 10). The influence of SST on the RF increases as the cloud becomes thicker. Additionally, when SST and COT increases, and consequently the infra-red radiation is trapped between surface and ice cloud, $RF_{tir}$ becomes larger. Finally, CER does not have as large as an effect on $RF_{tir}$ as it does in the SW wavelength range.





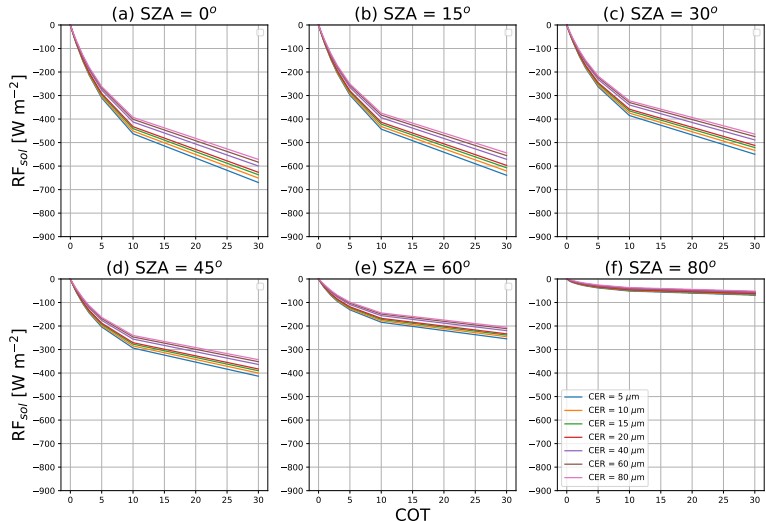

**Figure 2.** Radiative forcing (RF) in the shortwave (SW) wavelength range ($RF_{sol}$) as a function of the cloud optical thickness (COT) for different values of ice crystal effective radius (CER) for an ice cloud over the ocean shown for six different solar zenith angle (SZA) values ranging from **(a)** $0^o$ to **(b)** $80^o$.

Table 1 provides a summary of all the parameter used in RT simulations with their respective symbol, unit, and range of values.

## 2.4 Surface temperature and vertical temperature profiles

As will be seen in Section 3.1, an important input for the LW RT simulations is the surface temperature, called skin temperature (SKT), in the geographic area of interest for each selected day. SKT is defined as a radiative temperature of the model surface. SKT maps are downloaded from the Meteorological Archival and Retrieval System (MARS) archive for the forecast stream (fc) of the European Center for Medium-Range Weather Forecasts (ECMWF) for the six selected days. The data are re-gridded on a regular grid with a spatial resolution equal to $0.04^o$, covering the geographic domain of this study (see Figure 4). The data are based on the 00:00:00 UTC and 12:00:00 UTC analysis, each covering the subsequent 12-hour forecast period in a time resolution of one hour. For MSG/SEVIRI observations falling between two time steps, linear interpolation is performed between the two successive time steps to assign SKT maps to those observations.

To assess the validity of using a single atmospheric vertical temperature profile, radiative transfer simulations were performed for selected pixels by using real vertical temperature profiles instead of the LUTs in a number of randomly selected pixels (see Section 5.0.1). For this purpose, we downloaded hourly ERA5 vertical temperature profiles from the ECMWF's MARS archive for the analysis stream re-gridded onto a regular grid with a spatial resolution of $0.04^o$. The profiles are provided in





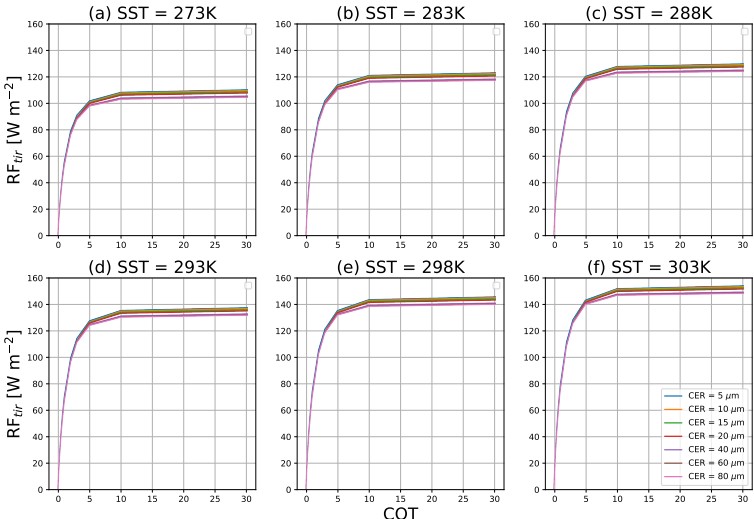

**Figure 3.** Radiative forcing (RF) in the longwave (LW) wavelength range ($RF_{tir}$) as a function of the cloud optical thickness (COT) for different values of ice crystal effective radius (CER) for an ice cloud over the ocean shown for six different sea surface temperature (SST) values ranging from **(a)** 273K to **(f)** 303K. For visual clarity, the SST value equal to 278K is not shown here.

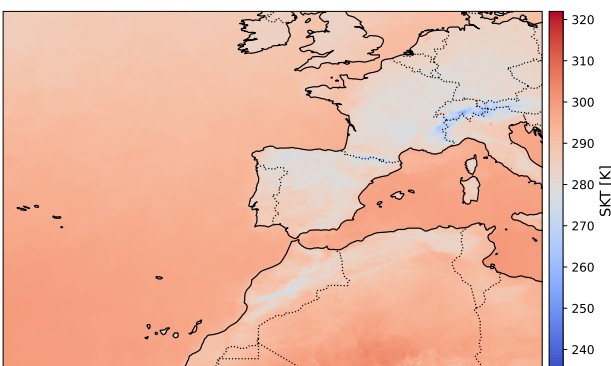

**Figure 4.** Skin temperature (SKT) map for an example date and observation time (25th of September 2023 and 06:00:00 UTC) over the study area.

model levels, which are 137 in total. We computed the pressure on these model levels and then, the geopotential height. The temperature profiles are temporally interpolated into the MSG/SEVIRI observation time by using linear interpolation between the analysis data at the previous and next hour.



**Table 1.** Parameter values used in the SW and LW radiative transfer model (RTM) simulations.

| Parameter | Symbol | Wavelength range | Units | Values |
|---|---|---|---|---|
| Ice cloud optical thickness | COT | SW/LW | 1 | 0, 0.1, 0.2, 0.3, 0.4, 0.5, 0.7, 1.0, 2.0, 3.0, 5.0, 10.0, 30.0 |
| Solar zenith angle | SZA | SW | degrees | 0, 5, 10, 15, 20, 25, 30, 35, 40, 45, 50, 55, 60, 65, 70, 75, 80 |
| Ice crystal effective radius | CER | SW/LW | $\mu$m | 5, 10, 15, 20, 40 , 60, 80 |
| Ice cloud top height | CTH | LW | km | 6, 7, 8, 9, 10, 11, 12, 13 |
| Sea surface temperature | SST | LW | K | 273, 278, 283, 288, 293, 298, 303 |
| Land surface temperature | LST | LW | K | 263, 268, 273, 278, 283, 288, 293, 298, 303, 308, 313 |
| Water cloud optical thickness | wCOT | SW/LW | 1 | 0.1, 0.5, 1.0, 5.0, 30.0 |
| Water cloud top height | wCTH | LW | km | 1, 2, 3, 4, 5 |

## 2.5 Land use

Using a representative land use dataset for the geographic area of interest is essential because it affects the RT simulations in the
LW wavelength range through the surface aldedo and emissivity. We utilize the Terra and Aqua combined Moderate-Resolution
Imaging Spectroradiometer (MODIS) Land Cover Type (MCD12Q1) Version 6.1 data for the most recent available year, 2022.
This data product provides global land cover types at yearly intervals derived from MODIS Terra and Aqua reflectance data
observations. The MCD12Q1 data are provided as tiles approximately $1000\text{x}1000km^2$ using a sinusoidal grid. In total, there
are 460 tiles covering the entire globe. The data are re-projected on a regular grid with a spatial resolution of $0.04^o$ in the
geographic domain of interest (see Fig. 5) with the aid of Satpy Python library (Hoese, 2019).

In this study, we employ the IGBP global vegetation classification scheme, which identifies 17 different land cover classes.
It should be noted that in the case of pixels being covered by multiple IGBP classes, we assign the land cover value with the
largest percentage coverage to that pixel, with a special treatment for water. We first discriminate whether the pixel is covered
by water for more than half of its fine-scale MODIS pixels, in which case it is assigned the water class. Else, we consider all
non-water classes for the majority choice.



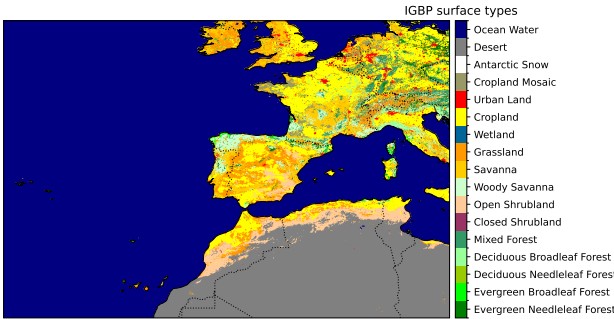

**Figure 5.** International Geosphere Biosphere Programme (IGBP) land use classification scheme in the selected geographic domain of this study from the MCD12Q1 Version 6.1 data product.

## 2.6 Polar Orbiting Satellite Observations

The CERES instruments aboard Terra measure the Earths' total radiation budget (Barkstrom, 1999). This type of satellite observations are essential for assessing the radiative effect of clouds.

The Single Scan Footprint (SSF) TOA/Surface Fluxes and Clouds product contains instantaneous CERES observations for a single scanner instrument (NASA, 2019). The data used in this study combine CERES observations with scene and cloud properties from MODIS on Terra and Aqua satellite. As the footprint of CERES (approximately 20 km) is larger than that of MODIS, the scene identification and cloud properties are averaged over the CERES footprint.

In this study, we use TOA SW and LW fluxes from CERES Terra- Flight Model 1 (FM1) and Aqua FM3 Edition 4A SSF to validate the TOA SW and LW flux maps of contrails derived using the Rapid Contrail-RF Estimation Approach (see Subsection 5).

## 3 Methodology of the Rapid Contrail-RF Estimation Approach

Here, we present the methodology followed, first to merge the three principal datasets introduced in Section 2 with the necessary additional data, and finally, to separate contrails from natural cirrus clouds (Subsection 3.2).

### 3.1 Merging the principal datasets

RF maps of contrails are generated by merging three principal datasets together with additional data.

First, as described in Section 2.1, we use the MSG/SEVIRI Dust/RGB composite to identify specific days during which contrails could be visually detected. For these particular days, the OCA product is re-gridded onto a regular grid with a spatial resolution equal to $0.04^o$, covering the study area, which expands from $30^oW$ to $15^oE$ longitude and $25^oN$ to $55^oN$ latitude. The necessary additional data, including SKT data (see Section 2.4 and the surface type (see Section 2.5), are also resampled onto the same regular grid with the same spatial resolution as the re-gridded OCA product. As a result, the generated maps





contain pixels with information on the cloud phase (cloud-free, liquid, ice, or multi-layered clouds), CTP, COT, CER, SKT, and surface type.

For each pixel characterized as an ice or multi-layered cloud, we first determine which LUT should be used among the three scene scenarios by utilizing the surface type information (land or ocean) and the OCA cloud phase (ice or multi-layered). Once the choice of LUT is made, a multi-dimensional interpolation of the simulated $RF_{sol}$ and $RF_{tir}$ values from the LUT at the actual values of the cloud and environmental parameters for each pixel (SZA, COT, SKT etc.) is performed.

The dimensions of the interpolation are determined by the number of parameters on which $RF_{sol}$ and $RF_{tir}$ depend in each LUT. For instance, for an ice cloud above ocean and within the SW wavelength range, a 3-dimensional interpolation is performed with the simulated $RF_{sol}$ being a function of COT, CER, and SZA parameters. For the same scene within the LW wavelength range, a 4-dimensional interpolation is necessary, where simulated $RF_{tir}$ is a function of COT, SST, CTH, and CER parameters.

The final output of this approach is the construction of $RF_{sol}$ and $RF_{tir}$, as well as $F_{sol}$ and $F_{tir}$ maps of the detected ice clouds.

### 3.2 Distinguishing natural and contrail cirrus clouds

Merging the different datasets, as explained in Section 3.1, will generate RF and fluxes maps of both natural and contrail cirrus clouds. To effectively distinguish them, we apply a filtering method based on the ice clouds' CTP value similar to the approach described in Wang et al. (2024).

The rationale behind this approach is that most of commercial airplanes fly at altitudes ranging from 8 to 12 km, which corresponds to a mean pressure level of 250 hPa. However, for a persistent contrail cirrus to form, specific atmospheric conditions are required, mainly the presence of an ISSR. In average, these ISSRs are typically found at slightly higher pressure levels, around 300 hPa. Combining both information, we implement a CTP filter at 300 hPa in our analysis, meaning that clouds above 300 hPa are considered to be contrails. It should be note that this approach provides a rough and approximate filtering compared to better approaches, such as manually labeling remote sensing images (Meijer et al., 2022), or contrail detection algorithms based on machine learning (Ortiz et al., 2024).

### 4 Results

In this Section, we first present the main results of the study (Subsection 4.1 and 4.2) including the detection and characterization of contrails as well as their RF. This is followed by four different validation exercises to assess the performance of the Rapid Contrail-RF Estimation Approach in generating realistic RF maps for contrails (Subsection 5).

### 4.1 Detection and characterization of contrails

The detection and characterization of contrails are conducted for each selected day by zooming on smaller geographic regions within our spatial domain of interest, based on the presence of a large number of contrails. This choice significantly reduces



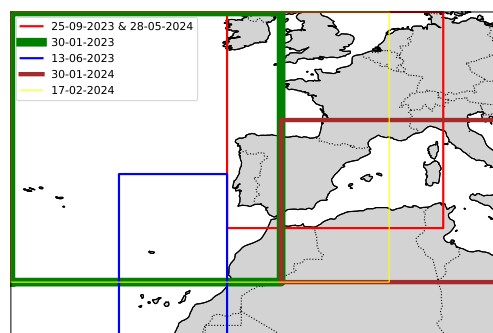

**Figure 6.** Geographic map of the overall geographic region, illustrating different colored boxes that represent the zoomed geographic areas of interest per day.

**Table 2.** Selected days, and the geographic limits of each zoomed regions within the geographic area of interest where persistent contrails were detected.

| Day | Longitude range | Latitude range |
|---|---|---|
| 30-01-2023 | [30$^o$W, 5$^o$W] | [30$^o$N, 55$^o$N] |
| 13-06-2023 | [20$^o$W, 10$^o$W] | [25$^o$N, 40$^o$N] |
| 25-09-2023 | [10$^o$W, 10$^o$E] | [35$^o$N, 55$^o$N] |
| 30-01-2024 | [5$^o$W, 15$^o$E] | [30$^o$N, 45$^o$N] |
| 17-02-2024 | [30$^o$W, 5$^o$E] | [30$^o$N, 55$^o$N] |
| 28-05-2024 | [10$^o$W, 10$^o$E] | [35$^o$N, 55$^o$N] |

the computational time for Sections 5.0.1, 5.0.3, and 5.1, while allowing us to sample pixels over land or ocean on the different days. Figure 6 shows the selected longitude and latitude ranges for each day, while Table 2 provides a summary.

Figure 7 shows an example of the DUST/RGB composite from SEVIRI/MSG, focusing on the geographic region over France and the Bay of Biscay for September 25th, 2023, from 06:00:00 UTC to 07:15:00 UTC. It should be noted that, for the sake of visual clarity, Figures 7, 8, and 9 demonstrate only a part of the zoomed geographic region where contrails are observed. At 06:00:00 UTC, a dense ice cloud is observed, surrounded by line-shaped ice clouds over the northern Bay of Biscay and France. As time progresses, the dense ice cloud mass disperses, while at the same time, we observe the formation of several line-shaped ice clouds around it.

As it is shown in Figure 8, the OCA algorithm successfully detects the dense ice cloud at 06:00:00 UTC as a mixture of ice and multi-layered clouds. Interestingly, as time progresses, most of the formed line-shaped contrails are characterized as



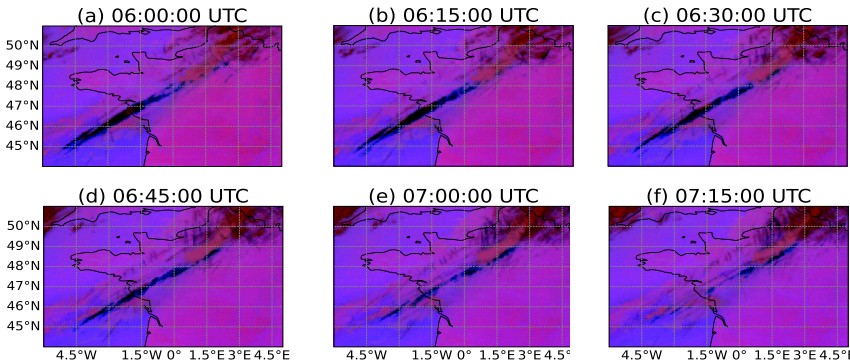

**Figure 7.** MSG/SEVIRI Dust/RGB images for an example date and sequence of observation times (25th September 2023) in a zoomed geographic area, where contrails are detected.

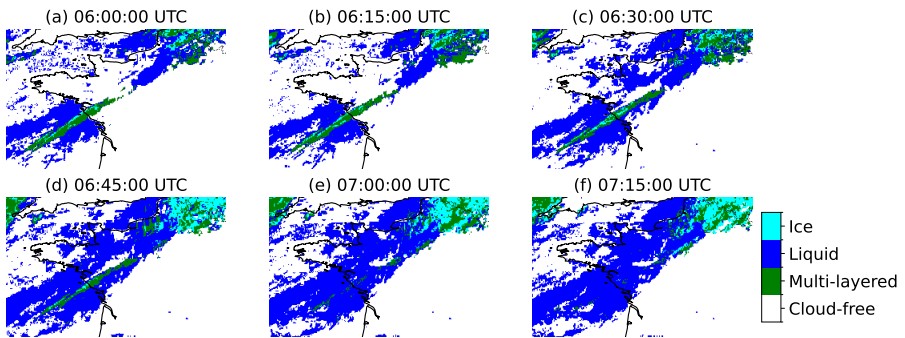

**Figure 8.** Cloud phase (cloud-free or multi-layered, liquid or ice cloud) as retrieved by the Optimal Cloud Analysis (OCA) algorithm for an example date and sequence of observation times (25th September 2023) in a zoomed geographic area where contrails are detected.

clouds by the OCA algorithm. We observe that many ice clouds are identified as clouds only in their central parts, while their thinner edges often remain undetected. We speculate that this is due to the spatial resolution of SEVIRI/MSG observations, which is 3x3 $km^2$ at the SSP.

330    As presented in Section 3.2, the application of a CTP filter distinguishes natural cirrus clouds from contrails. In the following section, we proceed with the generation of RF maps based on this distinction. Table 3 summarizes the average number of pixels characterized as ice and multi-layered clouds for each selected day per observation, as well as the percentage of these pixels having contrails. The day with the largest number of contrail pixels is the 17th of February 2024, followed by the 30th of January 2024.





**Table 3.** Selected days, average number of ice and multi-layered cloud pixels, and average number of contrail pixels per observation, and percentage of contrails over the overall geographic region.

| Day | Average number of ice and multi-layered cloud pixels | Average number of contrail pixels | Percentage of contrails |
|---|---|---|---|
| 30-01-2023 | 122.146 | 30.880 | 25.28 % |
| 13-06-2023 | 134.342 | 41.971 | 31.24 % |
| 25-09-2023 | 129.806 | 43.196 | 33.28 % |
| 30-01-2024 | 226.090 | 48.087 | 21.26 % |
| 17-02-2024 | 258.712 | 59.403 | 22.96 % |
| 28-05-2024 | 161.647 | 42.645 | 26.38 % |

## 4.2 Radiative forcing of contrails

Starting from this section and onward, the focus is exclusively on contrail cirrus clouds. A net RF value was assigned to the pixels characterized as ice or multi-layered clouds that passed the distinguishing filter between natural and contrail cirrus clouds.

The net RF is calculated as the sum of $RF_{sol} + RF_{tir}$. As discussed in Section 2.3, the presence of an ice cloud results in cooling in the SW wavelength region (negative RF values) and warming in the LW wavelength region (positive RF values).

Figure 9 shows maps of contrail net RF for the same geographic region as in Figures 7 and 8 on the 25th of September, 2023, from 06:00:00 UTC to 07:15:00 UTC. For this sequence of observation times, the detected contrails exhibit a positive radiative effect, indicating that the overall effect was warming during these early morning hours. At 06:00 UTC, the long, thin ice cloud mostly located above the Atlantic exhibits the strongest warming effect compared to other ice clouds during the rest of the time period, likely due to the still nighttime conditions in this region. As daytime progresses and the sun rises, the warming effect of the contrails diminishes, indicating that the shortwave cooling effect becomes more pronounced.

An overall daily view of the RF effect of contrails is provided in Figure 10. Over the geographic region of interest and for the six selected days, the net RF values of the detected ice or multi-layered cloud pixels have been multiplied by the coverage area per pixel, summed and then divided by the total coverage area for all the pixels. We refer to this summation as the total $RF_{all}$. Additionally, summing up only the contrail pixels will provide us the total $RF_{contrail}$. For the example day of 25th of September 2023 (Fig. 10 (c)), the total net RF values of the contrails range from -24.18 $W/m^2$ (12:00 UTC) to 31.99 $W/m^2$ (23:30 UTC). As it is expected, for all the selected days, the maximum total net $RF_{contrail}$ appears during nighttime due to the absence of SW cooling, while the minimum $RF_{contrail}$ value occurs during daytime and close to each midday. Even though the largest number of contrail pixels is found during the 17th of February 2024, we observe that the absolute maximum $RF_{contrail}$ values are observed during the 13th of June 2023 (see Table 3). This is due to the increased incoming solar radiation during the warmer months in the Northern Hemisphere compared to the colder months. Additionally, the contribution of the LW RF to the total RF for the detected ice and multi-layered clouds, and contrail pixels is shown in each subplot in Figure





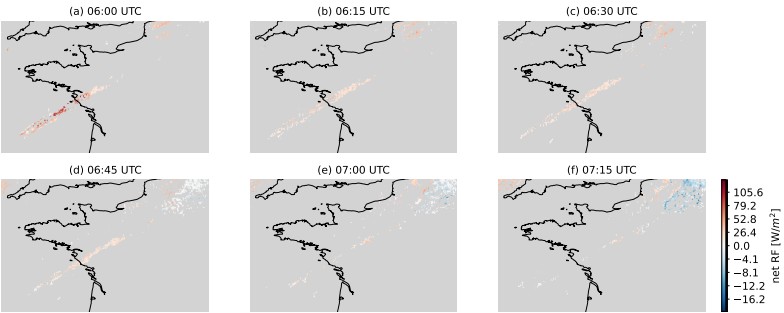

**Figure 9.** Net radiative forcing (RF) (sum of SW and LW RF) of contrails for an example date and sequence of observation times (25th September 2023) in a zoomed geographic area with detected contrails.

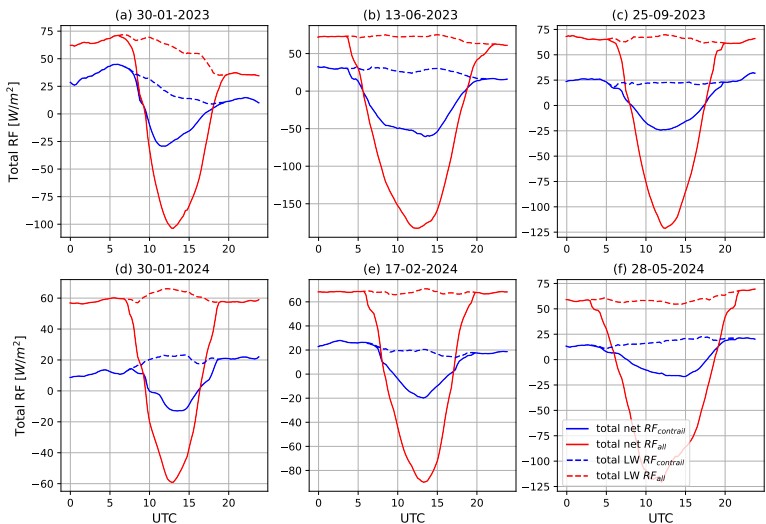

**Figure 10.** Time series of total net and longwave (LW) radiative forcing (RF) in $W/m^2$ for all the detected ice and multi-layered clouds (represented by solid and dashed red lines) and for only the detected as contrails (represented by solid and dashed blue line) above the overall geographic area for the six selected days.

10. It is observed that, for each case, the LW consistently contributes positively to the total RF throughout the day, with small fluctuations observed across different cases.



## 5 Validation of the Rapid Contrail-RF Estimation Approach


The accuracy and reliability of the Rapid Contrail-RF Estimation Approach in constructing RF maps for contrails have been investigated through four different validation exercises, presented in the following subsections. These exercises focus on different aspects of the methodology. First, we evaluate the choice of using a single atmospheric vertical profile in the RT simulations (Subsection 5.0.1). Next, by performing a small subset of RT simulations, we investigate the impact of selecting a certain ice

cloud parameterization scheme (Subsection 5.0.2). Additionally, we evaluate the impact of using CTH values estimated by a single atmospheric vertical profile on the RF estimations. Finally, we perform a comparison between the flux maps for contrails and polar-orbiting satellite observations (Subsection 5.1).

### 5.0.1 Impact of vertical temperature profile on radiative transfer calculations

The core component of the Rapid Contrail-RF Estimation Approach is the construction of the ice cloud RF LUTs and their

merging with the re-gridded geostationary maps (see Section 3.1). As presented in Section 2.3, the atmospheric temperature vertical profile used in the RT simulations remains constant and corresponds to the U.S. Standard Atmosphere.

To assess the validity of this choice and estimate the uncertainty associated with using a single constant temperature vertical profile, randomly selected pixels from the zoomed geographic regions of each day-containing contrails above land, ocean, and water clouds (i.e., multi-layered)-covering day- and night-time conditions were chosen as the sample of this investigation.

For these selected pixels, RT simulations were performed using the ERA5 vertical temperature profile from ECMWF (see Section 2.4) as the input atmospheric profile. These profiles were also used to estimate CTH and wCTH (only in the presence of a water cloud). Additionally, for each pixel, the actual CER and COT values from the OCA product were used, along with the real SZA. In the presence of a water cloud, we use the wCOT value from the OCA product.

In Figure 11, for each scene scenario, we present the comparison results between the RF values coming from the LUTs

($RF_{USstandard}$) and the RF values calculated by using the actual atmospheric and cloud conditions ($RF_{ERA5}$) per selected pixel in the SW and LW wavelength ranges, separately. As it can be seen, for all the scene scenarios in the SW wavelength range, overall good agreement is found with the correlation coefficient and slope values being close to unity, with the exception of a few comparison points. Table 4 provides some statistics for the two different methodologies followed in this Section per wavelength and scene scenario. In the SW wavelength range, the use of LUTs instead of real-time RT simulations per pixel can

lead to RMS error equal to $6.13\ W/m^2$, $10.76\ W/m^2$, and $11.99\ W/m^2$ above land, ocean, and water cloud, respectively. The comparisons in the LW wavelength range (see Figure 11) reveal an overall good agreement with correlation coefficient values being around 1.00 and slope values in the range of 0.95 - 0.97. In contrast to the comparison in the SW, in the LW, we observe that a larger number of points appears to be scattered around the 1:1 line. This finding means that the RT simulations in the LW wavelength range are more sensitive in the choice of the atmospheric temperature vertical profile. The use of LUTs in the LW

wavelength range leads to RMS error values of the same order of magnitude for the three scene scenarios. When focusing on the SW and LW RMS error percentage, we find that the largest values for both wavelength ranges are observed for the scene scenario of an ice cloud above a water cloud (multi-layered).



To explain the scattered points around the 1:1 line in the subplots of Figure 11, we focus on the points with an RMS error value larger than the mean RMS error value plus two times the standard deviation of the RMS error. For these points, we first investigated whether there is a correlation between the large discrepancies in the two RF datasets and the differences between the values of each actual cloud parameter and the closest values used during the multi-dimensional interpolations in the LUTs. The comparison results showed no correlation.

Additionally, for these points, we examine the corresponding ECMWF vertical profiles used in the RTM simulations. Figure 12 illustrates the temperature and humidity of the US Standard profile, along with the median profile of the ECMWF vertical profiles, as well as the coverage of these profiles. We observe that the coverage of the ECMWF vertical profiles shows different values for surface temperatures but their median profile agrees very well with the US Standard atmospheric profile. In contrast, the humidity ECMWF vertical profiles show a large difference at the surface compared to the US Standard profile.

**Table 4.** Mean radiative forcing (RF) values over all the randomly selected pixels for the six selected days, bias, RMS error, RMS error percentage, and mean percent errors between RF values estimated by using the Look-Up Tables (LUTs) and by using the ERA5 atmospheric profile and the OCA cloud conditions for the SW, and LW estimated RFs.

| | Mean RF value (USstandard) ($W/m^2$) | Mean RF value (ERA5) ($W/m^2$) | Bias ($W/m^2$) | RMS Error ($W/m^2$) | RMS Error percentage (%) | Mean bias percentage (%) |
|---|---|---|---|---|---|---|
| Land/ SW | -95.28 | -97.27 | 1.99 | 6.13 | 6.30 | 2.05 |
| Multi-layered/ SW | -71.21 | -68.28 | -2.93 | 11.99 | 17.56 | 4.29 |
| Ocean/ SW | -145.49 | -146.97 | 1.48 | 10.76 | 7.32 | 1.01 |
| Land/ LW | 84.46 | 86.74 | -2.29 | 7.53 | 8.68 | 2.64 |
| Multi-layered LW | 61.35 | 65.23 | -3.88 | 7.01 | 10.75 | 5.95 |
| Ocean/ LW | 95.32 | 98.19 | -2.88 | 7.24 | 7.37 | 2.93 |

Overall, in the SW wavelength range, the use of a standard profile in the construction of the LUTs lead to mean bias percentage of about 2.05%, 1.01%, and 4.29% for a contrail above land, ocean, and water cloud, respectively. In the LW wavelength range, the mean percent errors equal to 2.64%, 2.93%, and 5.95% for a contrail above land, ocean, and water cloud, respectively.





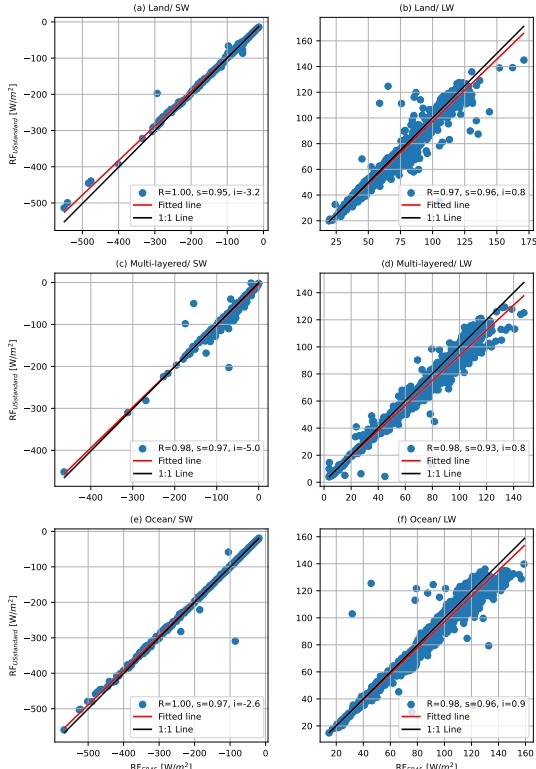

**Figure 11.** Scatter plot between radiative forcing (RF) values estimated by using the Look-Up Tables (LUTs) ($RF_{LUTs}$) and radiative transfer calculations using the actual atmospheric temperature vertical profiles ($RF_{actual}$) for randomly selected pixels containing contrails above land surfaces in the (a) SW, and (b) LW, underlying water clouds (i.e., multi-layered) in the (c) SW and (d) LW, and ocean surfaces in the (e) SW and (f) LW.

### 5.0.2 Impact of ice cloud parameterization on radiative transfer calculations

The micro-physical properties of the ice crystals, which are part of the cirrus clouds and contrails, play a crucial role in their single scattering properties and, consequently, the RF of these clouds (Stephens et al., 1990; Sanz-Morère et al., 2020). Here, we assess the impact related to the choice of ice cloud parameterization in the RT simulations. The parameterization determines how the ice water content and CER are translated into optical properties. Since the ice crystal shape is an unknown parameter, we have selected the parameterization by Yang et al. (2013), assuming the ice crystal habit to be a column composed of 8 elements with a moderate degree of roughness, as this is the habit most frequently observed for thin ice clouds (Forster and




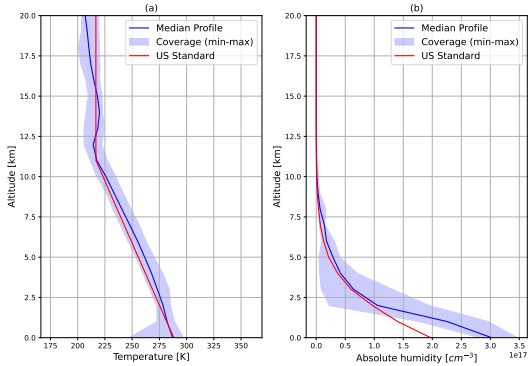

**Figure 12.** Vertical (a) temperature and (b) humidity profiles of US Standard atmosphere, median profiles of the ECMWF vertical profiles corresponding to the largest discrepancies (i.e., large RMS error percentage between radiative forcing (RF) values estimated by using the Look-Up Tables (LUTs) ($RF_{LUTs}$) and radiative transfer calculations using the actual atmospheric temperature vertical profiles) for the six selected days.

Mayer, 2022). According to the same study, 60 % of cirrus clouds are a mixture of ice crystals with severe roughness, while
40 % a mixture of smoothed ones. Similarly to Wolf et al. (2023), we have chosen a moderate degree of roughness for the simulations included in the LUTs.

For this sensitivity study, we performed a small subset of RT simulations in the SW and LW wavelength ranges, varying the choice of ice cloud parameterization. We selected all the available ice crystal shapes from the parameterization by Yang et al. (2013). In addition, we included the parameterization by Fu (1996); Fu et al. (1998), which is operationally applied in
the ECMWF Integrated Forecasting System (IFS) and assumes ice crystals as pristine hexagonal columns. The simulations are always performed for an ice cloud with a COT equal to 0.5 to maximize its semi-transparency and, subsequently, the effect of cloud microphysics. We have chosen three different SZA values ($10^o$, $40^o$, and $70^o$), a CER of 20 $\mu m$, and a CTH of 10 km. For these simulations, the ice cloud is located above an ocean surface characterized by three different SST values (273 K, 293 K, and 303 K).

Figure 13 shows $RF_{sol}$ as a function of various ice crystal habits based on the parameterization of Yang et al. (2013) (i.e., column with 8 elements, droxtal, hollow bullet rosette, hollow column, plate, plate with 10 elements, plate with 5 elements, solid bullet rosette, and solid column) and their degrees of roughness (smooth, moderate, and severe) for three different SZAs. The ice crystal habit of an hexagonal column by Fu (1996) is included as well. Additionally, the figure presents the relative differences in $RF_{sol}$ compared to the selected ice crystal shape and degree of roughness for the construction of the LUTs. As
observed, the choice of ice crystal habit and roughness degree can result in large differences, which can be up to 60% (e.g., the case for SZA = $10^o$ for smooth plates of 10 elements) in the SW wavelength range. In addition, the parameterization of Fu (1996), which assumes a pristine hexagonal column results in differences up to approximately 20 % for the case of a small



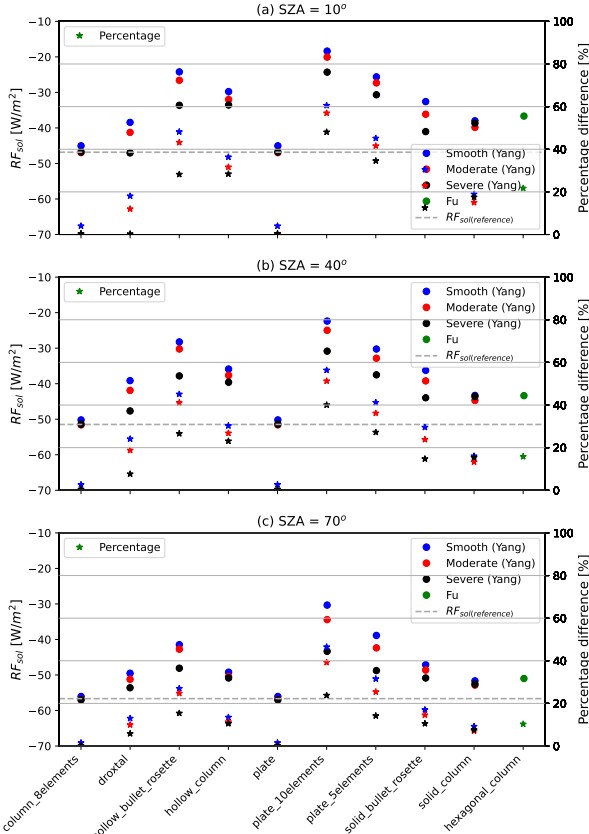

**Figure 13.** Simulated radiative forcing values in the shortwave (i.e., solar) wavelength range ($RF_{sol}$) are shown as a function of various ice crystal habits and their degrees of roughness based on the parameterization of Yang et al. (2013) and Fu et al. (1998) for three different solar zenith angle (SZA) scenarios. The horizontal line (i.e., grey dashed line) represents the $RF_{sol(reference)}$ value for the selected ice crystal shape and roughness used in this study.

SZA. For the three SZA scenarios, $RF_{sol}$ of the selected ice crystal shape and roughness appears to have the lowest values compared to other ice crystal shapes and degrees of roughness.

Figure 14 shows $RF_{tir}$ as a function of the same ice crystal habits and roughness degrees for three different SST scenarios, along with their relative differences. In contrast to the shortwave range, the differences in the LW ($RF_{tir}$) are much smaller, not exceeding 12%.

From the sensitivity tests, we conclude that ice crystal habit and roughness can lead to significant differences in RT simulations in the SW wavelength region, while these factors play a less significant role in the LW wavelength region. When

investigating the simulated upward and downward irradiance at TOA in the SW wavelength region, we find that the largest




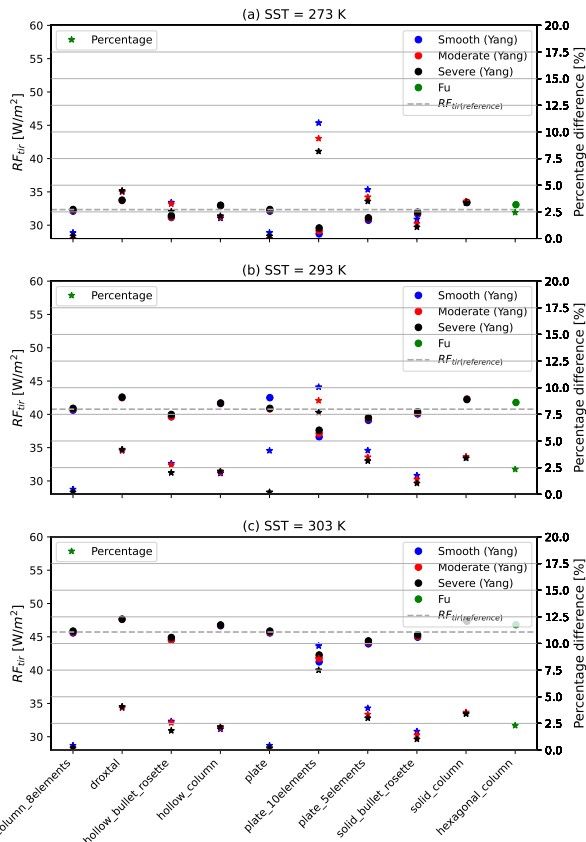

**Figure 14.** Similar as Figure 13 but for the longwave (i.e., thermal infrared) wavelength range, where $RF_{tir}$ values are shown for three different sea surface temperature (SST) scenarios.

differences between the selected ice crystal shape and roughness (i.e., column of 8 elements with moderate roughness) and a plate of 10 elements with a smooth degree of roughness (i.e., largest differences in $RF_{sol}$) occur in the following wavelength ranges: 1122 - 1135 nm, 1346 - 1471 nm, 1800 - 1954 nm, and 2486 - 2752 nm. Similarly, in the LW wavelength, the simulated spectrum is affected the most by the choice of the ice crystal shape and roughness in the following wavelength ranges: 3487 -

4171 nm, 4645 - 5502 nm, and 8113 - 9153 nm.

To estimate the uncertainty associated with the selection of a specific ice cloud parameterization in the $RF_{sol}$ and $RF_{tir}$ maps, we have re-performed the RT simulations for the randomly selected pixels (see Section 5.0.1 for only a single day; 25-09-2023). Consequently, the comparison is made between $RF_{sol}$ and $RF_{tir}$ obtained in Section 5.0.1, where the default ice cloud parameterization was applied, and those generated by employing the same input values for the RT simulations, but

differing the choice of ice cloud parameterization. For the comparison, we have used the ice crystal habit and roughness, which



exhibits the largest difference with our default settings: plate of 10 elements with a smooth degree of roughness (Yang et al., 2013).

Table 5 summarizes the findings of the above-mentioned comparison. As expected by the sensitivity study, the use of another ice crystal habit and roughness can lead to large differences in the SW and slightly affects the LW wavelength range. For the

SW wavelength range and for all the scene scenarios, the mean RF values for columnar and plate ice crystals differ by a negative bias, with the largest bias found for contrails above ocean surfaces (-49.33 $W/m^2$).

For the LW wavelength range, the bias values are smaller, with the largest bias being equal to 5.89 $W/m^2$ for ice clouds above water clouds (i.e., multi-layered).

We should keep in mind that actual measurements of the micro-physical properties of ice crystals in contrail clouds are rare

and difficult to obtain. There have been in-situ measurements, such as those in Järvinen et al. (2018), which found that the primary ice crystal habit is aggregates (i.e., the one used in this study), though the presence of other crystal shapes has been reported. Consequently, we used the most common one to optimize the representation of ice crystals. However, applying a single ice crystal shape and roughness for the overall number of detected contrails during different seasons, and above various scenes may not be fully representative.

**Table 5.** Mean radiative forcing (RF) values over all the randomly selected pixels for the 25th of September 2023, bias, RMS error between RF values when using an ice crystal habit of column with 8 elements and a plate with 10 elements for the SW, LW, and net estimated RFs.

| | Mean RF value (column 8elements) ($W/m^2$) | Mean RF value (plate 10elements) ($W/m^2$) | Bias ($W/m^2$) | RMS Error ($W/m^2$) |
|---|---|---|---|---|
| Land/ SW | -77.14 | -42.70 | -34.44 | 37.36 |
| Multi-layered/ SW | -66.65 | -34.39 | -32.26 | 41.57 |
| Ocean/ SW | -133.52 | -84.19 | -49.33 | 60.08 |
| Land/ LW | 63.20 | 58.37 | 4.83 | 5.24 |
| Multi-layered LW | 47.08 | 41.19 | 5.89 | 7.13 |
| Ocean/ LW | 65.55 | 67.29 | -1.75 | 15.26 |

### 5.0.3 Impact of Cloud Top Height (CTH) on radiative forcing interpolation

As mentioned in Section 2.2, the OCA product provides the CTP. To have the information about CTH, which is used as a parameter in the RT calculations in the LW wavelength range, the US Standard profile is used. More precisely, we linearly interpolate the CTP in the pressure vertical grid, and consequently, the altitude vertical grid, of the US Standard atmospheric profile.

CTH plays an important role in the LW wavelength range, where it is utilized to perform the multi-dimensional interpolation of the simulated $RF_{tir}$ values from the LUT at the actual values of the cloud and environmental parameters for each pixel.




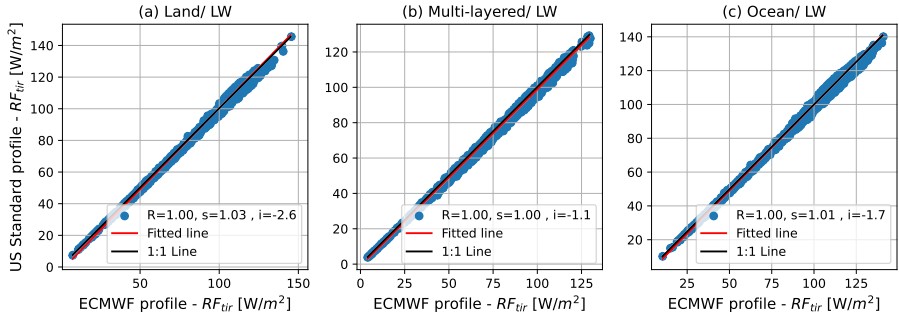

**Figure 15.** Scatter plots between radiative forcing (RF) values estimated by using the cloud top height (CTH) as estimated by the US Standard profile and by the ECMWF temperature profiles for randomly selected pixels containing a contrail above (a) land surfaces, (b) underlying water clouds and (c) ocean surfaces in the LW wavelength range.

To evaluate the accuracy of using a constant atmospheric profile to estimate the CTH, we have selected the same pixels as in Section 5.0.1. For these pixels, ECMWF pressure vertical profiles were used to interpolate linearly the contrail CTP in the altitude vertical grid of those profiles. Consequently, this CTH value, named 'CTH - ECMWF', in every selected pixel can be used to re-perform the multi-dimensional interpolation in the parameters and estimate a new RF value in the LW wavelength range.

Figure 15 shows the comparisons between this new RF value by using the actual CTH and a CTH estimated by the US Standard profile. As we can see, the correlation between them is excellent for the three different scene scenarios, indicating that using a different CTH value does not affect the multi-dimensional interpolation performed in the LUTs to extract the $RF_{tir}$. As it is shown in Figure 16, for the three different scene scenarios, the CTH values estimated by using a real atmospheric and the US Standard profile show small differences with a mean bias equal to 0.85 %, -0.60 %, and -1.70 % above land, ocean, and water cloud, respectively. The scatter plots of Figure 16 reveal that for the three scene scenarios, CTH estimated by the US Standard profile is systematically lower by 22 - 26 % compared to the CTH estimated by the ECMWF vertical profiles.

## 5.1 Comparison of estimated flux maps and CERES observations

TOA upward solar (i.e., SW) and thermal infrared (i.e., LW) fluxes, as observed by the CERES FM1 and FM3 instruments, have been used to validate the first output after merging the datasets in the Rapid Contrail-RF Estimation Approach: the TOA upward SW and LW fluxes referred to as $F_{up}$ (see Equation 1). This comparison focuses exclusively on pixels identified as contrail pixels.

The comparison was conducted using data from the six selected days. For each of these days, the closest-in-time MSG/-SEVIRI observation was matched with the CERES observations (approximately four per day above the zoomed geographic region of interest) by taking into account the exact acquisition time of the selected MSG/SEVIRI pixels. Since CERES has a larger footprint (approximately 25 km in diameter near nadir) compared to the spatial resolution of the flux maps generated by the Rapid Contrail-RF Estimation Approach, we averaged the contrail cirrus pixels, which are located inside the CERES



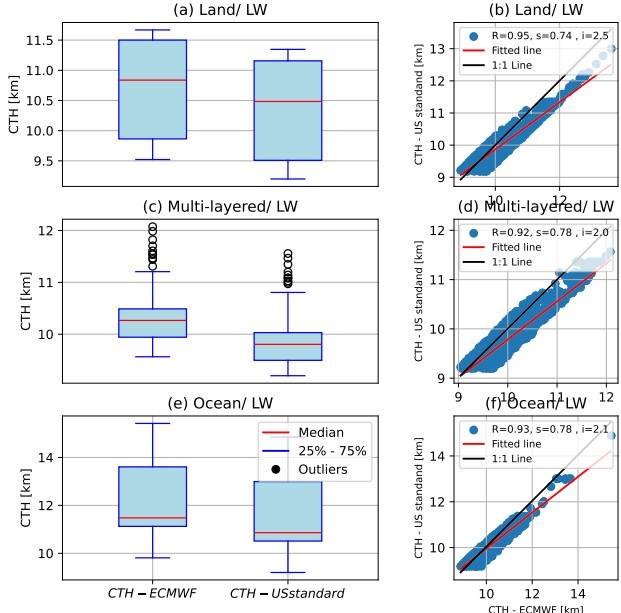

**Figure 16.** Box-Whisker plots of cloud top height (CTH) values as estimated by the US Standard profile and by actual temperature profiles from ECMWF for randomly selected pixels containing a contrail above (a) land surfaces, (b) underlying water clouds and (c) ocean surfaces in the LW wavelength range.

footprint. To perform this averaging, we defined a circular area with a radius of 12.5 km around the latitude and longitude of
CERES field-of-view (FOV) at surface. If this area was fully covered of contrail pixels, we averaged these pixels and compared them with the CERES fluxes in the SW and LW wavelength ranges, separately.

Figure 17 shows the outcome of the comparison described above. As seen, there is generally good agreement in the SW and LW wavelength range, with a correlation coefficient (R) of 0.90 and 0.85, respectively.

Table 6 provides some overall statistics for the mean CERES and Rapid Contrail-RF Estimation-estimated upwards TOA
fluxes. The bias in the SW wavelength range (6.92 $W/m^2$) indicates that the Rapid Contrail-RF Estimation Approach, in general, slightly overestimates the SW fluxes by 2.33% compared to CERES, while in the LW wavelength range the bias is negative (-12.20 $W/m^2$) indicating that our approach underestimates in general the LW fluxes by 7.67 % compared to CERES. Additionally, the higher Rapid Contrail-RF Estimation Approach SW fluxes compared to CERES align with the $RF_{sol}$ values for the ice cloud microphysics in Section 5.0.2. There, it was shown that using column of 8 elements as the ice crystal shape
results in the lowest radiative forcing values compared to other ice crystal shapes. Concerning the RMS error, the Rapid Contrail-RF Estimation Approach seems to perform better in estimating LW fluxes than SW ones. However, we observe that in LW wavelength range, the two compared fluxes exhibit the largest scattering, resulting in many points having considerably lower fluxes compared to CERES ones.



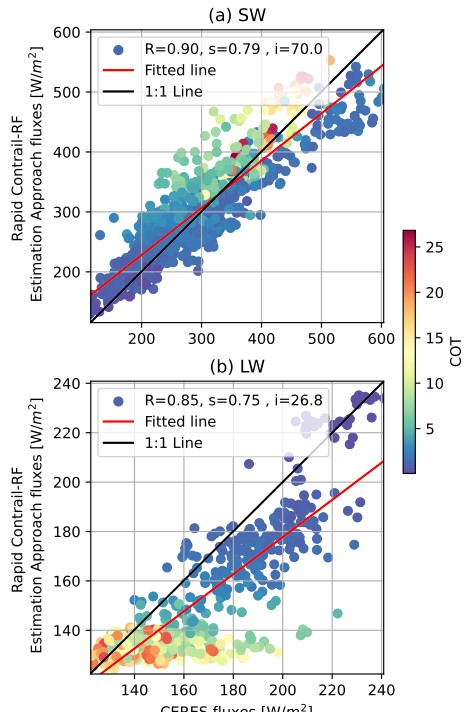

**Figure 17.** Scatter plots of TOA fluxes observed by CERES and estimated by using the LUTs in the shortwave (SW, upper plot) and longwave (LW, lower plot) wavelength ranges. The points of each plot are color-coded based on the mean contrail optical thickness (COT for cloud optical thickness), averaged within the respective CERES footprints.

Focusing on the mean contrail optical thickness (COT) values, averaged within the respective CERES footprints and color-coded in Figure 17, we observe that in the LW wavelength region, the largest COT values correspond to cases in which the Rapid Contrail-RF Estimation Approach retrieves low fluxes (smaller than 140 $W/m^2$), while the CERES fluxes show larger variability. In the SW wavelength, there is no direct correlation between mean contrail optical thickness values and discrepancies between the two datasets.





**Table 6.** Mean CERES and Rapid Contrail-RF Estimation Approach-estimated upwards TOA fluxes, bias and RMS error for the SW and LW wavelength range.

| Wavelength range | Mean CERES fluxes $(W/m^2)$ | Mean Rapid Contrail-RF Estimation fluxes $(W/m^2)$ | Bias $(W/m^2)$ | RMS Error $(W/m^2)$ |
|:---:|:---:|:---:|:---:|:---:|
| SW | 296.60 | 303.52 | 6.92 | 44.12 |
| LW | 159.03 | 146.82 | -12.20 | 18.33 |

## 6 Conclusions

Quantifying the radiative forcing of aviation-induced clouds and contrails remains an active area of research, primarily due to significant uncertainties surrounding their overall contribution to climate change. In this study, a new satellite-based contrail radiative forcing mapping is presented and evaluated. The so-called Rapid Contrail-RF Estimation Approach combines geostationary satellite observations, a cloud properties retrieval algorithm, radiative transfer modeling and a simplistic separation scheme between natural and contrail cirrus clouds.

For six selected days within the 2023-2024 period, during which contrails were visually identified, MSG/SEVIRI data in combination with the OCA product were used for the detection and characterization of contrail cirrus clouds and aviation-induced cloudiness. The central focus of this study is the application of pre-computed RF LUTs for thin to semi-transparent ice clouds in both SW and LW spectral regions. SW and LW RF values were assigned to pixels identified as ice clouds using a multi-dimensional interpolation scheme. This methodology is computationally fast, avoiding the need for real-time radiative

transfer simulations, and enabling the processing of large datasets.

The primary aim of this study was to evaluate the validity and limitations of using RF LUTs in this context. This evaluation was conducted through four different validation exercises. The first three focused on (1) the choice of using a single atmospheric vertical profile for the LUTs construction, (2) the choice of using one single ice cloud parameterization scheme and finally, and (3) the impact of using CTH values estimated with a standard profile during the merging of the cloud product with the LUTs.

The fourth validation exercise is an end-to-end validation, comparing contrail flux maps generated by the Rapid Contrail-RF Estimation Approach with those derived from CERES instruments.

The main findings of the first three validation exercises are as follows:

1. Using a single standard atmospheric profile - in this study, the US Standard atmospheric profile - for constructing RF LUTs generally provides promising results. Indeed, this assumption can introduce biases of up to -2.93 $W/m^2$ and -3.88

$W/m^2$ in the SW and LW wavelength range, respectively, which are considerably smaller compared to the respective fluxes.





2. Using a single ice crystal habit - in this study, a column composed of 8 elements with a moderate degree of roughness - for constructing the RF LUTs can lead to significant differences in the SW wavelength region. In the LW wavelength region, RF values are less sensitive to this selection. Using the most extreme difference scenario, which does not necessarily reflect reality, the choice of ice crystal habit can result to biases of up to -49.33 $W/m^2$ and 5.89 $W/m^2$ in the SW and LW wavelength range, respectively.

3. Using a CTH estimated from a single standard atmospheric profile during the merging of the cloud product with the RF LUTs leads to small differences in the LW wavelength range (i.e., biases of up to -1.70 $W/m^2$).

The end-to-end validation, which compared contrail flux maps generated by the Rapid Contrail-RF Estimation Approach with those derived from CERES instruments, yielded encouraging results concerning the performance of the Rapid Contrail-RF Estimation Approach. The mean biases are found to be 6.92 $W/m^2$ and -12.20 $W/m^2$ for the SW and LW wavelength ranges, respectively. The observed biases can be partially attributed to the selected ice crystal habit, as the chosen habit tends to produce the lowest RF values compared to the other options.

Averaging all the mean biases percentage from the different correlative comparison in the LW and SW wavelength ranges, we find that our approach provides accurate data for estimating contrail radiative forcing, with an accuracy on the order of approximately 15 %.

The resulting contrail RF maps revealed that, for the six selected days in this study, the presence of contrails causes warming during nighttime and cooling during daytime. The total daily mean net RF values caused by contrails over the entire geographic area of this study were calculated as follows: 6.91 $W/m^2$ (25-09-2023), 3.05 $W/m^2$ (28-05-2024), 11.11 $W/m^2$ (30-01-2023), -12.09 $W/m^2$ (13-06-2023), 8.23 $W/m^2$ (30-01-2024), and 9.93 $W/m^2$ (17-02-2024). During the only summer month included in the analysis, the total daily mean net RF value is negative indicating that the SW contribution to the net RF in larger than the LW contribution. This is due to the increased incoming solar radiation during the warmer months in the Northern Hemisphere compared to the colder months.

To conclude, our study presents a new satellite-based contrail radiative forcing mapping. Performing various validation exercises, we demonstrate that this method provides reliable SW, LW and net RF maps for contrail cirrus clouds. Based on these findings, future steps could include extending this study to cover a full year, which we believe will offer valuable insights into the seasonal behavior of contrails. Furthermore, leveraging more advanced geostationary satellites with higher spatial and temporal resolution, such as Meteosat Third Generation/ Flexible Combined Instrument (MTG/FCI) would contribute in a better detection and monitoring of contrails. Finally, implementing an improved separation scheme between contrails and natural cirrus clouds - such as a contrail detection algorithm based on neural networks as proposed by Ortiz et al. (2025) - would enhance the detection of contrails.

*Code and data availability.* We have published the code to prepare an IGBP map and an example usage of the Look-UP Tables. Additionally, we provide the IGBP map on the reference domain, the NetCDF file for the look-up tables, and the timing of the SEVIRI images. This material has been published on Zenodo: https://zenodo.org/records/14859250



*Author contributions.* ED designed the validation of the Rapid Contrail-RF Estimation Approach, conducted the experiments, collected various datasets, performed the RT simulations, merged the principal datasets, carried out the data analysis, and wrote this paper. PdB contributed to preparing the land use dataset, providing the CERES observations, and developing the merging process of the principal datasets to generate the RF maps. NC conceived the initial idea for the validation of the Rapid Contrail-RF Estimation Approach and initiated the design of the LUTs. All authors contributed and reviewed the final paper.

*Competing interests.* The contact author has declared that none of the authors has any competing interests.

*Acknowledgements.* This research work is part of the E-CONTRAIL project (https://www.econtrail.com/), funded from the SESAR Joint Undertaking (JU) under grant agreement No. 101114795. The authors gratefully acknowledge the E-CONTRAIL partners and the SESAR Joint Undertaking for their support, fruitful discussions during project meeting, and funding the E-CONTRAIL project. We would like to thank all the people who have contributed to the development of libRadtran software. Additionally, we would like to thank the Atmospheric
Science Data Center at NASA Langley Research Center for providing the CERES data, as well as EUMETSAT for providing the MSG/SE-VIRI and OCA data. Finally, we would like to thank Dr. Christine Aebi for proofreading this manuscript.



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
