# Peer review of "Satellite-based estimation of contrail cirrus cloud radiative forcing derived through a Rapid Contrail-RF Estimation Approach"

_EGUsphere, 2025_

## Referee Comment (RC2)

Manuscript number: egusphere-2025-697
Full title: Satellite-based estimation of contrail cirrus cloud radiative forcing derived through a Rapid Contrail-RF Estimation Approach
Author(s): Dimitropoulou E. et al.

The authors developed a rapid system for evaluating contrail radiative forcing using geostationary satellite observations, a contrail cirrus detection algorithm, and look-up tables constructed from a radiative transfer model. The authors analyzed several days of datasets that include detected contrail cirrus clouds to estimate the shortwave, longwave, and net radiative effects of contrail cirrus and validate the resultant estimations with those from CERES products, showing the accuracy of the estimated "contrail" radiative forcing to be about 15%. The present paper shows novel results regarding the contrail radiative forcing estimation with careful uncertainty evaluations and quantification. The topic in the present paper is suitable for *Atmospheric Measurement Techniques (AMT)*. However, the present contrail RF estimation relies on the accuracy of the contrail detection algorithm, and the authors do not describe the performance evaluation of the detection algorithm. This manuscript requires major revisions before reconsideration of publication. Please find the specific comments below.

**Major comments**
1. **Uncertainty of the contrail cirrus detection and separation methods:** The algorithms for the contrail detection and separation from natural cirrus clouds would be among the key factors that impact the contrail RF estimations. However, I do not see any description regarding the accuracy of the present contrail detection/separation method (e.g., Page 5, Lines 124-127). In addition, the descriptions in Lines 305-307 let me suspect the possibility of misdetection of natural cirrus clouds as contrail cirrus because the authors applied a very simple CTP filter to detect contrails. Finally, this suspicion became more confident with the scatter plot of COTs in Fig. 17 that shows COTs of contrail cirrus clouds to be optically thicker than ~3 for most of the cases and thicker than 20 for some cases. I do not think that these are all from contrail cirrus clouds. Although the authors show a contrail cirrus case in Figs. 7-8, it is not sufficient to inform of the accuracy of the contrail detection method. Without a solid validation of the contrail detection/separation method, the representativeness of the quantitative estimation of the "contrail" RF is questionable. The authors should at least provide quantitative discussions of the contrail detection/separation methods or cite the reference that describes a comprehensive validation of the method.
2. **Discussion of contrail RF does not contrast with previous studies.** Previous studies have also evaluated contrail RF, but the present study did not compare their estimations with those estimated through the previous studies. The authors should discuss the results with the previous studies in terms of contrail RF [1-3] and optical properties [4-5]. I suspect that the estimated contrail RF in the present study is too large compared to these previous studies, even if cloud fractions are taken into account.

**Minor comments**
1. Page 1, Line 17 "Wielicki et al. (1995)": This should be "(Wielicki et al., 1995)". This type of error is seen throughout the manuscript, and I suggest the authors double-check the reference format.

2. Page 2, Line 29: Use parentheses for the references.
3. Page 2, Lines 33: "later properties of the persistent contrail cirrus clouds" sounds a bit odd. Probably, rephrase it with "the properties of the resultant persistent contrail cirrus clouds."
4. Page 5, Line 133 "DEKOUTSIDIS and FEIDAS": Use lower cases for the reference.
5. Page 5, Lines 136-138: Ill-posed problem is the condition that the measurements do not have sufficient information to unambiguously estimate the state vector. I suggest the authors improve the descriptions.
6. Page 12, Lines 281-282: What is the lower detection limit of contrails (in terms of COT)?
7. Page 153 Line 307: "note" should be "noted"
8. Page 14, Line 317: Please check the required format for the subsection. Should the subsection start from 0?
9. Page 16, Line 345 "likely due to the still nighttime conditions in this region": This will become unambiguous if the authors co-plot SZA along UTC in Fig. 10.
10. Figure 13: I suggest the authors take off the legends in (a-b) as they overlap with plots and are redundant. Also, the caption needs to describe what the symbols (i.e., circles and stars) indicate.
11. Page 26, Line 499 "some": This should be deleted.
12. Page 28, Lines 533-534: I think that the statement should be restricted to the region of interest (i.e., Europe) and suggest the authors revise the corresponding description to be "…provides promising results over the region of interest."

**Reference**

[1] Kärcher, B. (2018). Formation and radiative forcing of contrail cirrus. *Nature communications*, *9*(1), 1824.
[2] Burkhardt, U., & Kärcher, B. (2011). Global radiative forcing from contrail cirrus. *Nature climate change*, *1*(1), 54-58.
[3] Chen, C. C., & Gettelman, A. (2013). Simulated radiative forcing from contrails and contrail cirrus. *Atmospheric Chemistry and Physics*, *13*(24), 12525-12536.
[4] Iwabuchi, H., Yang, P., Liou, K. N., & Minnis, P. (2012). Physical and optical properties of persistent contrails: Climatology and interpretation. *Journal of Geophysical Research: Atmospheres*, *117*(D6).
[5] Schumann, U., Baumann, R., Baumgardner, D., Bedka, S. T., Duda, D. P., Freudenthaler, V., ... & Wang, Z. (2017). Properties of individual contrails: a compilation of observations and some comparisons. *Atmospheric Chemistry and Physics*, *17*(1), 403-438.

---

## Author Response (AR1)

Dear Editor, Anonymous Referee #1, and Anonymous Referee #2,

We would like to sincerely thank you for your valuable feedback, comments, and suggestions.

Please find attached our revised manuscript, as well as the author's version showing track changes relative to the initially submitted manuscript. We have also provided our detailed responses to each referee in the interactive discussion of our AMT preprint.

For your convenience, we again provide a point-by-point response to the reviewers' comments, highlighting the corresponding changes made in the manuscript.

We reply to each of the Referee's comments individually below and indicate the corresponding modifications/additions made to the manuscript.

Please consider that:
→ Black:
    Comments of the Referee
→ Black italic:
    The response to each comment posed by the Referee.
→ **Black bold:**
    **Already existing text in the manuscript.**
→ **Red bold:**
    **Added text in the manuscript according to the Referee's comment.**

**Response to Anonymous Referee #1:**

The authors developed a rapid system for evaluating contrail radiative forcing using geostationary satellite observations, a contrail cirrus detection algorithm, and look-up tables constructed from a radiative transfer model. The authors analyzed several days of datasets that include detected contrail cirrus clouds to estimate the shortwave, longwave, and net radiative effects of contrail cirrus and validate the resultant estimations with those from CERES products, showing the accuracy of the estimated "contrail" radiative forcing to be about 15%. The present paper shows novel results regarding the contrail radiative forcing estimation with careful uncertainty evaluations and quantification. The topic in the present paper is suitable for Atmospheric Measurement Techniques (AMT). However, the present contrail RF estimation relies on the accuracy of the contrail detection algorithm, and the authors do not describe the performance evaluation of the detection algorithm. This manuscript requires major revisions before reconsideration of publication. Please find the specific comments below.

**Response:**
*We would like to thank the Referee for the valuable feedback provided on our manuscript entitled "Satellite-based estimation of contrail cirrus cloud radiative forcing derived through a Rapid Contrail-RF Estimation Approach". The comments and suggestions will help us improve the overall quality of the manuscript.*

**Major comments**
**1. Uncertainty of the contrail cirrus detection and separation methods:** The algorithms for the contrail detection and separation from natural cirrus clouds would be among the key factors that impact the contrail RF estimations. However, I do not see any description regarding the accuracy of the present contrail detection/separation method (e.g., Page 5, Lines 124-127). In addition, the descriptions in Lines 305-307 let me suspect the possibility of misdetection of natural cirrus clouds as contrail cirrus because the authors applied a very simple CTP filter to detect contrails. Finally,

this suspicion became more confident with the scatter plot of COTs in Fig. 17 that shows COTs of contrail cirrus clouds to be optically thicker than ~3 for most of the cases and thicker than 20 for some cases. I do not think that these are all from contrail cirrus clouds. Although the authors show a contrail cirrus case in Figs. 7-8, it is not sufficient to inform of the accuracy of the contrail detection method. Without a solid validation of the contrail detection/separation method, the representativeness of the quantitative estimation of the "contrail" RF is questionable. The authors should at least provide quantitative discussions of the contrail detection/separation methods or cite the reference that describes a comprehensive validation of the method.

**Response:**

*In the first submission of the manuscript, we did not point out explicitly that the cloud top pressure (CTP) filter that we use in this work is intended as a simplification for forthcoming work with collaborators who specialize on this topic. The scope of the present work lies in the radiative forcing (RF) estimation method, which should be combined with a contrail detection method for producing meaningful contrail impact studies. The issue of contrail detection was also raised by Reviewer 2, and we agree that the CTP filter does not allow for proper contrail detection. To improve the message and the correctness of our manuscript, we have thus taken the following actions:*

1. *Clarify the role of the CTP filter as a simplification due to the focus of our work on RF estimation methods.*
2. *Explain the different types of contrails (contrails, persistent contrails, cirrus contrails) and which of those are amenable to a study by geostationary imagers. Our method is intended to work on ice clouds in general, of which contrails are a subset.*

*The major changes have been made in the Introduction (Lines 40 - 42, Lines 77 - 81, and Lines 95 - 97) and in Section 2.3 and are too extensive to reproduce here. We also added a note in the Conclusions about the combination of proper contrail detection with our method being a necessary followup work (Lines 596 - 598).*

**2. Discussion of contrail RF does not contrast with previous studies.** Previous studies have also evaluated contrail RF, but the present study did not compare their estimations with those estimated through the previous studies. The authors should discuss the results with the previous studies in terms of contrail RF [1-3] and optical properties [4-5]. I suspect that the estimated contrail RF in the present study is too large compared to these previous studies, even if cloud fractions are taken into account.

**Response:**
*Reviewer 1 raises important issues, which we classify as follows: (i) the role of the contrail detection, (ii) the optical properties of the clouds in our data, (iii) the representativeness of contrails in our results, and (iv) discussion of our results with previous studies.*

*(i) Role and uncertainty of the contrail detection. This issue was raised by reviewer 2 as well. In the present manuscript, we present a RF estimation method is applicable to "ice clouds" in general. Our goal is to present and validate the method, using a substitute for contrail detection (the cloud top pressure filter). This method was also used by Wang et al. (2023), which allows for a direct comparison. The submitted manuscript was not consistent and explicit about this choice and we have edited to manuscript to make this distinction clear in the Introduction but also when describing the method.*

*(ii) Optical properties of the clouds. In this work, we rely on the Optimal Cloud Analysis (OCA) product of EUMETSAT. This is an operational purpose retrieval products for the geostationary instrument that provides day and night characterization of the clouds. It is generated and*

*disseminated by EUMETSAT as the reference cloud retrieval product for the SEVIRI instrument. We decided to use it as it is a reference usable by others and because it has been validated. The product guide for OCA-SEVIRI mentions that improvements are necessary for the characterization of cirrus clouds: the main limitation is the detection of thin clouds over a thick underlying cloud. Those improvements would directly benefit users of the Rapid Contrail RF method.*

*(iii) Our elements of response indicate that we cannot, based solely on the CTP filter, discriminate on the role of contrails versus cirrus clouds in our results. What we can comment on, however, is the selectivity of geostationary imagers to contrails. In Driver et al (2025), the authors compare simulated imagery of contrails, based on CoCiP simulations, with the direct physical output of CoCiP. There, they find that contrail detection is limited at low and high optical thickness. The result of contrail detection by geostationary imager is thus always limited in range and this should be taken into account in future work.*

*(iv) Concerning the discussion of our results with previous studies, we agree that providing a comparison with other studies enhances the scientific relevance of our work and clarifies its added value. In the revised manuscript, we have therefore included Section 5.5 entitled "Comparison with an existing study", where we compare the SW, LW, and net RF for our Rapid Contrail-RF Approach with those reported by Wang et al. (2024 for two consecutive contrail outbreaks over Western Europe. We selected this specific comparison because both studies rely on the same geostationary satellite observations and the same cloud product (considering that Wang et al. (2024) apply some modifications). However, the RF estimation methods differ, allowing us to directly assess the performance of our approach relative to theirs.*

*Other studies in the literature (e.g., Burkhardt et al., 2011; Chen et al., 2013; Bock et al., 2016; Yi et al., 2012) generally report global and/or regional results from models, typically over one-year periods. In contrast, our current analysis, focuses on a 6-day period to evaluate the performance of the Rapid Contrail-RF Estimation Approach. We believe that even a qualitative comparison between our daily mean net RF values and annual or monthly mean values from these studies would not be meaningful and could be misleading.*

*As part of our ongoing research, we are currently working on extending the analysis to a 1-year dataset on a larger European region.*

*References*

- Burkhardt, U., & Kärcher, B. (2011). Global radiative forcing from contrail cirrus. *Nature climate change, 1*(1), 54-58.
- Chen, C. C., & Gettelman, A. (2013). Simulated radiative forcing from contrails and contrail cirrus. *Atmospheric Chemistry and Physics, 13*(24), 12525-12536.
- Bock, L., & Burkhardt, U. (2016). Reassessing properties and radiative forcing of contrail cirrus using a climate model. *Journal of Geophysical Research: Atmospheres, 121*(16), 9717-9736.
- Yi, B., Yang, P., Liou, K. N., Minnis, P., & Penner, J. E. (2012). Simulation of the global contrail radiative forcing: A sensitivity analysis. *Geophysical Research Letters, 39*(24).
- EUMETSAT: Document EUM/TSS/MAN/14/770106 "Optimal Cloud Analysis: Product Guide"

**Minor comments**
**1. Page 1, Line 17 "Wielicki et al. (1995)": This should be "(Wielicki et al., 1995)". This type of error is seen throughout the manuscript, and I suggest the authors double-check the reference format.**
**Response:**
*We agree with the Referee's comment and have corrected the text accordingly. All the references are checked throughout the manuscript and corrected. Additionally, we have updated the full*

*reference list, including DOIs and ensured that journal names are abbreviated according to the Journal Title Abbreviations by Caltech Library.*

**2. Page 2, Line 29: Use parentheses for the references.**
**Response:**
*Please consider our response to the previous comment.*

**3. Page 2, Lines 33: "later properties of the persistent contrail cirrus clouds" sounds a bit odd. Probably, rephrase it with "the properties of the resultant persistent contrail cirrus clouds."**
**Response:**
*We have modified this sentence as follows:*
**Lines 39 - 40: "...ice crystal number, will affect the properties of the resulting contrail cirrus clouds (Unterstrasser, 2016)."**

**4. Page 5, Line 133 "DEKOUTSIDIS and FEIDAS": Use lower cases for the reference.**
**Response:**
*Please consider our response to Comment #1 from the minor comments.*

**5. Page 5, Lines 136-138: Ill-posed problem is the condition that the measurements do not have sufficient information to unambiguously estimate the state vector. I suggest the authors improve the descriptions.**
**Response:**
*We agree with the Referee. Following Comment #45 of Referee #2 (Technical corrections), we have shortened Chapter 2, and this section has been removed from the manuscript.*

**6. Page 12, Lines 281-282: What is the lower detection limit of contrails (in terms of COT)?**
**Response:**

*According to Driver et al. (2025), the lower detection limit of contrails in terms of COT varies depending on contrail width. However, regardless of width, contrails with a COT below approximately 0.05 are undetectable. This finding is consistent with Karcher et al. (2009).*
*The OCA algorithm applied to MSG/SEVIRI data typically sets the lower limit for a reliable retrieval of a thin cloud at around 0.3-0.5 COT, though this depends on the surface type and viewing geometry. Even when this lower limit is applied, the retrieval uncertainty, which indicates the quality of the measurement, can be high.*
*We have added this information in the manuscript as:*
**Line 282: "It should be noted that contrails with COT values lower than 0.05 are undetectable when using imaging instruments aboard geostationary satellites (Karcher et al., 2009; Driver et al., 2025)."**

**References:**
- Driver, O. G. A., Stettler, M. E. J., and Gryspeerdt, E.: Factors limiting contrail detection in satellite imagery, Atmos. Meas. Tech., 18, 1115–1134, https://doi.org/10.5194/amt-18 1115-2025, 2025.
- Kärcher, B., Burkhardt, U., Unterstrasser, S., and Minnis, P.: Factors controlling contrail cirrus optical depth, Atmos. Chem. Phys., 9, 6229–6254, https://doi.org/10.5194/acp-9-6229-2009, 2009.

**7. Page 153 Line 307: "note" should be "noted"**
**Response:**
*We have incorporated this change.*

**8. Page 14, Line 317: Please check the required format for the subsection. Should the subsection start from 0?**
**Response:**
*The comment is correct and resulted from a mislabeling of the subsection as a sub-subsection.*
*The labels have been corrected accordingly.*

**9. Page 16, Line 345 "likely due to the still nighttime conditions in this region": This will become unambiguous if the authors co-plot SZA along UTC in Fig. 10.**
**Response:**
*We have modified Figure 10 by adding a shaded grey area to indicate nighttime conditions, making it visually easier to identify these periods. Furthermore, the RF values have been revised after correcting a bug in our code. We apologize for this earlier error.*

**10. Figure 13: I suggest the authors take off the legends in (a-b) as they overlap with plots and are redundant. Also, the caption needs to describe what the symbols (i.e., circles and stars) indicate.**
**Response:**
*We have incorporated these changes in Figure 13 and 14 as well as in Figure 13's caption.*

**11. Page 26, Line 499 "some": This should be deleted.**
**Response:**
*We have incorporated this change.*

**12. Page 28, Lines 533-534: I think that the statement should be restricted to the region of interest (i.e., Europe) and suggest the authors revise the corresponding description to be "...provides promising results over the region of interest."**
**Response:**
*We have incorporated this change.*

References

[1] Kärcher, B. (2018). Formation and radiative forcing of contrail cirrus. Nature communications, 9(1), 1824.
[2] Burkhardt, U., & Kärcher, B. (2011). Global radiative forcing from contrail cirrus. Nature climate change, 1(1), 54-58.
[3] Chen, C. C., & Gettelman, A. (2013). Simulated radiative forcing from contrails and contrail cirrus. Atmospheric Chemistry and Physics, 13(24), 12525-12536.
[4] Iwabuchi, H., Yang, P., Liou, K. N., & Minnis, P. (2012). Physical and optical properties of persistent contrails: Climatology and interpretation. Journal of Geophysical Research: Atmospheres, 117(D6).
[5] Schumann, U., Baumann, R., Baumgardner, D., Bedka, S. T., Duda, D. P., Freudenthaler, V., ... & Wang, Z. (2017). Properties of individual contrails: a compilation of observations and some comparisons. Atmospheric Chemistry and Physics, 17(1), 403-438.

**Response to Anonymous Referee #2:**

**General comments**

3.  In this work, the authors developed a method to estimate the radiative forcing of contrail cirrus using geostationary satellite data, from SEVIRI on MSG. Their method is based on

detecting contrails and contrail cirrus on the satellite images, applying an algorithm to retrieve the cloud properties and using precomputed radiative transfer look-up tables (LUTs). A rather very simplified separation technique between natural and contrail cirrus is applied to isolate the RF from contrail cirrus. The authors perform an extensive evaluation of their method by performing four distinct tests: they assessed the use of a single vertical profile for the construction of their LUTs, the use of a single ice habit and the use of the cloud top height estimated from a standard profile. Finally, they directly compared the findings from their method, to contrail RF maps derived from CERES.

The topics discussed in this paper are in the scope of AMT and the interest of its readers. Overall, despite some strong simplification applied in the method, the paper demonstrates a viable and efficient approach, which could most certainly be applied also for large-scale, near-real-time estimation of contrail cirrus RF, with potential applications in global climatologies. Nevertheless, a list of mostly minor and technical revisions is presented in the following.

**Response:**
*We would like to thank the Referee for their constructive review of our preprint entitled "Satellite-based estimation of contrail cirrus cloud radiative forcing derived through a Rapid Contrail-RF Estimation Approach." We are pleased to hear that the manuscript is of interest to the AMT readers and that our approach is considered a viable and efficient method for estimating contrails radiative forcing.*

**Specific comments**

- General comment throughout the text. Be more careful in addressing the studied clouds. Between contrails and contrail cirrus there are significant differences in micro- and macrophysical as well as radiative properties. Already in the abstract in the first sentence you study "the radiative forcing of contrail cirrus" by line 6 you "quantify the radiative effects of ice clouds" and in lines 9-10 "contrails cause a cooling effect". Arguably in your method some things are applied on ice clouds, some on contrails and some on contrail cirrus, but please be more precise about when you are talking about what since they are not always interchangeable. (Lines 32-34 explain exactly that).

**Response:**
*We thank the Reviewer for this important comment. We acknowledge that in the manuscript we often used the term "contrail" as a simplification, even though contrail and contrail cirrus differ in their microphysical, macrophysical, and radiative properties. We fully agree that these distinctions should be made explicit. To address this, we have carefully revised the text to improve consistency and precision in terminology.*

*In the manuscript, we consider persistent contrails (see classification below). One important reason is that young contrails are too thin to be observed by a multispectral imager. Even for persistent contrails, cloud retrieval methods likely miss a number of them. We refer to "Factors limiting contrail detection in satellite imagery" (Driver et al., 2025), where a combination of contrail simulation and simulated satellite imagery/detection indicates a lower bound of about 0.2-0.3 (COT) for the detection. The method in that reference also shows an upper bound of about 20-30 in COT for the detection. This type of limitation will be shared by any study that relies on similar instruments.*

*Apart from nomenclature, we also make the context for using the method more explicit by adding the following:*

**Lines 163 - 164:** *Apart for the range of the parameters, which should be verified by users of these tables, the method here is applicable to naturally occurring cirrus clouds as well as to contrails. In this work, we perform the validation on all ice clouds that are selected by a cloud top pressure filter.*

*The motivation for the method is however more specific: in subsequent work we will apply our method to contrail that have been detected using state-of-the-art Machine Learning methods for cloud detection. See "Enhancing GOES-16 Contrail Segmentation through Ensemble Neural Network Modeling and Optical Flow Corrections" (Ortiz et al., 2025) for more details. We thus consider this part to extend beyond the scope of the present manuscript.*

*Regarding the class of clouds that we consider, we have modified the following parts of the manuscript:*
*a) In the Abstract (Line 1), we now clearly distinguish between contrails (young, line-shaped ice clouds forming directly behind aircraft) and contrail cirrus (the persistent, evolved stage with broader spatial and temporal extent).*
*b) In the Introduction (Lines 34–43), we have clarified the physical processes leading from contrail formation to their possible transition into contrail cirrus, including their persistence in ice-supersaturated regions and their average lifetimes.*
*c) We have revised the occurrences of "contrail" and "contrail cirrus" throughout the manuscript. However, variables such as $RF_{contrail}$ and the name of our approach (e.g., Rapid Contrail-RF Estimation Approach) remained the same.*

- **Lines 30-31:** Admittedly I am not familiar with this publication, but contrails consist of ice crystals so it is ice saturation that is needed. Based on the saturation vapor pressures ice saturation is reached before liquid. Maybe another reference would be more suitable in this place.

**Response:**
*We have revised the sentence to clarify this points and replaced the previous reference with the following:*
**Line 34-37: These aviation-induced clouds are formed behind aircraft cruising in sufficiently cold air due to the emission of water vapor. If the ambient air is sufficiently humid (that is, the relative humidity with respect to ice exceeds 100%), the contrails can persist, as the ice particles within the contrails grow by deposition of water vapor molecules from the ambient air (Schumann, 2005).**

- **Lines 35-36:** That is a very bold statement, especially to be left uncited. Please reconsider, rephrase and cite to support.

**Response:**
*We agree with the Referee that the statement "These persistent contrails are the only ones relevant for changes in the Earth's radiation budget" is too strong and was not supported by a citation. Therefore, we have removed this sentence from the manuscript. We believe that the new paragraph starting with "The impact of…" - Lines 44 – 58 - provides a more appropriate and detailed explanation of the influence of contrails on the TOA radiation budget. This is also related to our reply to the next comment.*

- **Lines 39-40:** A bit more explanation on the shortwave and longwave radiation and their interactions with contrail cirrus could be useful. The cooling and warming effects are a

result of this interaction and they depend on the microphysical and optical properties of each individual contrail cirrus. References should be added on the interaction of clouds with short and longwave radiation as well as the resulting cooling and warming effects.

**Response:**
*We have taken this comment into account, and Lines 39-40 have been replaced with a more detailed paragraph that includes additional explanation and is supported by relevant citations. Please find the revised text in Lines 48-58 of the manuscript:*
***"Under most conditions, in the solar wavelength range (i.e., shortwave/SW), persistent contrails and contrail cirrus reflect incoming sunlight back to space, resulting in a negative radiative effect and thus a cooling influence. In the thermal-infrared wavelength range (i.e. longwave/LW), they trap outgoing LW radiation within the Earth-atmosphere system, leading to a positive radiative forcing of LW and an associated warming effect (Heintzenberg and Charlson, 2009). By adding both radiative components, the net radiative effect of the cirrus clouds and consequently, contrails, can be calculated. This net effect can be either positive or negative, depending on the microphysical, macrophysical and optical properties of the contrail cirrus, as well as the radiative properties of the environment (Wolf et al., 2023). For example, cloud properties such as the cloud optical thickness, cloud temperature, and ice crystal shape influence the net radiative response (Kärcher and Burkhardt, 2013; Stephens et al., 2004; Markowicz and Witek, 2011), while environmental parameters like the surface albedo and surface temperature can play a significant role (Schumann and Mayer, 2017)."***

- **Line 152:** Is that accurate?

**Response:**

*In the OCA ATBD, the 2-layer detection process is described in detail. Specifically, during situations with a high measurement cost function, the algorithm tests for a 2-layer condition. The initialization procedure included the following guidance:"Set FG, AP, and APerror variables according to the following table (with empirically determined example values given here for the meteorological 2-layer case, **ice above water cloud)**".*
*The default assumption in OCA for two-layer situations is thus to have an ice cloud over a water cloud. Although this assumption is not universally confirmed for every case, it reflects the most typical configuration.*

- **Lines 300-309:** I think I consider this one the weakest points in this study. For a dataset of 6 scenes even manual tracking would be sufficient. Contrail detection and tracking algorithms are also readily available.

**Response:**
*We agree with the reviewer that the method we have chosen is not amenable to the specific detection of contrails in general (see also our reply to "Specific comment 1"). We have consciously made the choice to not use an automated detection algorithm in this manuscript. We believe that it is a reasonable choice for three reasons:*

*(i) We present a RF estimation method that is applicable to "ice clouds" in general. We can thus apply it based on the clouds classified by OCA as "ice".*

*(ii) We selected small regions with contrail outbreaks so that our example still specializes on contrails.*

*(iii) This choice allows for a direct comparison with Wang et al. (2024), in which the authors also apply a CTP filter to a contrail outbreak.*

*Adding a dedicated contrail detection algorithm would not change our results but would make our work more complex and would involve extra parameters in the analysis of the results. It remains true that the radiative impact of contrails depends on both the detection and the RF estimate, but we believe that the techniques and the validations each deserve their own dedicated research works.*

*We have clarified the role of the CTP filter in the introduction and made explicit the limitation in the conclusion as well:*
**Lines 596 – 598:** *(…) implementing an improved separation scheme between contrails and naturally occurring ice clouds –such as contrail detection algorithms based on neural networks (Ortiz et al., 2025)– is a necessary further step to perform radiation forcing studies for aviation-induced cloudiness.*

*We are currently committed to a research activity in which we combine state-of-the-art machine learning based contrail detection methods, for which another institution is in charge, with our RF estimation method.*

- **Line 413:** Is it also the most common habit in contrail cirrus?

**Response:**
*According to Jarvinen et al. (2018), where cloud chamber studies of simulated cirrus clouds and globally distributed measurements of five airborne in-situ measurement campaigns targeting cirrus and contrails were exploited, the ice crystal shape that best matches the measurements is that of severely roughened column aggregates. In our study, we have chosen a moderate roughness; however, as shown in Figures 13 and 14, the differences in RFsol and RFtir between both roughness are minor. Please consider our modification in the manuscript:*
**Line 419: "the habit most frequently observed for thin ice clouds (Forster and Mayer, 2022) and contrails (Järvinen et al., 2018). "**

- **Chapter 5:** It would be preferable to also provide a comparison to other studies and methods in order to solidify the scientific importance of the presented work and its added value.

**Response:**
*We agree that providing a comparison with other studies enhances the scientific relevance of our work and clarifies its added value.*
*In the revised manuscript, we have therefore included Section 5.5 entitled "Comparison with an existing study", where we compare the SW, LW, and net RF for our Rapid Contrail-RF Approach with those reported by Wang et al. (2024 for two consecutive contrail outbreaks over Western Europe. We selected this specific comparison because both studies rely on the same geostationary satellite observations and the same cloud product (considering that Wang et al. (2024) apply some modifications). However, the RF estimation methods differ, allowing us to directly assess the performance of our approach relative to theirs.*
*Other studies in the literature (e.g., Burkhardt et al., 2011; Chen et al., 2013; Bock et al., 2016; Yi et al., 2012) generally report global and/or regional results from models, typically over one-year periods. In contrast, our current analysis, focuses on a 6-day period to evaluate the performance of the Rapid Contrail-RF Estimation Approach. We believe that even a qualitative comparison between our daily mean net RF values and annual or monthly mean values from these studies would not be meaningful and could be misleading.*
*As part of our ongoing research, we are currently working on extending the analysis to a 1-year dataset on a larger European region.*

**References**

- Burkhardt, U., & Kärcher, B. (2011). Global radiative forcing from contrail cirrus. *Nature climate change, 1*(1), 54-58.
- Chen, C. C., & Gettelman, A. (2013). Simulated radiative forcing from contrails and contrail cirrus. *Atmospheric Chemistry and Physics, 13*(24), 12525-12536.
- Bock, L., & Burkhardt, U. (2016). Reassessing properties and radiative forcing of contrail cirrus using a climate model. *Journal of Geophysical Research: Atmospheres, 121*(16), 9717-9736.
- Yi, B., Yang, P., Liou, K. N., Minnis, P., & Penner, J. E. (2012). Simulation of the global contrail radiative forcing: A sensitivity analysis. *Geophysical Research Letters, 39*(24).

**Technical corrections**

- **Abstract: First half of the abstract feels a bit incoherent. My preference would be a slightly longer abstract, but with proper structure and connections between sentences. Preferably the first sentence of the abstract should define the scientific gap and goals of the study, then followed by (a not too technical) introductions of the methodology and then results/evaluation. For example: "Contrail cirrus, anthropogenic clouds formed by cruising aircraft, strongly influence the Earth's radiation budget, but their exact Radiative Forcing (RF) remains poorly quantified at high temporal resolution. In this study we present a Rapid Contrail-RF Estimation Approach using geostationary satellite observations to estimate their radiative forcing. More precisely, observations from the Spinning Enhanced Visible and InfraRed Imager (SEVIRI) were utilized…**

**Response:**

*We agree with the Referees' comment and have modified the Abstract (Lines 1 - 18) as follows:*

"Contrails, anthropogenic ice clouds formed by aircraft at cruise altitudes, strongly influence the Earth's radiation budget but the measurement of their radiative forcing (RF) remains poorly quantified at high temporal resolution. In this study, we present the Rapid Contrail-RF Estimation Approach, which uses geostationary satellite observations to estimate their radiative forcing. Starting from a cloud retrieval product, we apply pre-computed Look-Up Tables (LUTs) to generate radiative forcing maps for natural and contrail cirrus clouds. Specifically, observations from the Spinning Enhanced Visible and InfraRed Imager (SEVIRI) onboard Meteosat Second Generation (MSG) were used to visually identify days with contrails. For six selected days, ice clouds were characterized using the Optimal Cloud Analysis (OCA) product from MSG/SEVIRI data provided by the European Organization for the Exploitation of Meteorological Satellites (EUMETSAT). The LUTs were constructed using the libRadtran radiative transfer model to quantify the radiative effect of ice clouds in the short-wave (SW) and long-wave (LW) spectral regions. A cloud top pressure filter was applied to isolate potential contrails. The resulting data set provides a quantification of SW, LW, and net radiative forcing at the top of the atmosphere due to potential contrails. We show that these clouds contribute to daytime cooling and nighttime warming, with a net effect that varies between diurnal cycles and seasons. We assess the validity of the Rapid Contrail-RF Estimation Approach through correlation exercises focusing on uncertainties in the use of LUTs, a single ice cloud parameterization, and a calculated cloud top height, supplemented by comparisons with polar orbiting satellite observations from the Clouds and the Earth's Radiant Energy System (CERES) instruments. In general, these correlative comparisons indicate that the proposed approach provides accurate data on the estimation of the radiative forcing of potential contrails, with an accuracy of approximately 15 %."

- **Line 9: Emphasize the finding. "Over the full diurnal cycle, contrails cause a cooling effect during the daytime and warming at night.". Proposed change: "Results show that contrails contribute to daytime cooling and nighttime warming, with a net effect that varies across diurnal cycles and seasons."**

**Response:**
*We have accepted the proposed change and have updated the abstract accordingly. This change has been implemented in the new version of the Abstract as mentioned in our response to the Technical corrections (comment no.1).*

- **Line 10: Have a cooling effect or cause cooling**

**Response:**
*We have accepted the proposed change and have updated the abstract accordingly. This change has been implemented in the new version of the Abstract as mentioned in our response to the Technical corrections (comment no.1).*

- **Line 14: data on radiative forcing estimation of contrails, with**

**Response:**
*We have accepted the proposed change and have updated the abstract accordingly. This change has been implemented in the new version of the Abstract as mentioned in our response to the Technical corrections (comment no.1).*

- **Line 17: crucial for mitigating climate change (Wielicki et al., 1995).**

**Response:**
*We agree with the Referee's comment and have corrected the text accordingly:*
**Line 20: "… for mitigating climate change (Wielicki et al., 1995)."**

- **Line 18: The latest IPCC report (add year or citation), highlights, that**

**Response:**
*We agree with the Referee's suggestion. We have revised the sentence and now reads as follows:*
**Line 21: "The IPCC Sixth Assessment Report (IPCC, 2023) highlights that…"**
*with the corresponding reference:*
**IPCC, ed.: Climate Change 2023: Synthesis Report. Contribution of Working Groups I, II and III to the Sixth Assessment Report of the Intergovernmental Panel on Climate Change, IPCC, Geneva, Switzerland, available at: https://www.ipcc.ch/report/sixth-assessment-report-synthesis-report/, 2023.**

- **Line 21: pollutants, being the two main**

**Response:**
*We have incorporated this change.*

- **Line 25: non CO2 effects**

**Response:**
*We have incorporated this change.*

- **Line 21 – 26: The structure is a bit confusing. Best to start by stating that the CO2 effects where the first ones to be clearly recognized which lead to them being also the focus of academic interest.**

**Response:**

*We agree with the Referee's suggestion. We have revised the paragraph and now reads as follows:*

**Line 24 – 30:**

**"The first effects to be clearly identified and linked to the observed global warming were those of CO2 emissions (Letcher, 2020), which is why many studies and reports initially focused on the quantification of aviation's contribution to the global atmospheric CO2 concentrations (Olsthoorn, 2001; Pejovic et al., 2008; Ji-Cheng and Yu-Qing, 2012; Mayor and Tol, 2010; Howitt et al., 2011). The delayed onset on research of the non-CO2 effects is not due to their insignificance for the climate, but rather because these effects are not yet fully understood and remain associated with considerable uncertainty (Lee et al., 2021)."**

- **Lines 27 – 28 : The non-CO2 aviation effects include emissions of pollutants, such as, nitrogen oxides (NOx = NO + NO2), water vapor (H2O), soot and sulfur oxides (SOx) as well as the formation of contrail cirrus clouds (Lee et al., 2021)**

**Response:**

*We have incorporated these changes.*

- **Lines 28 – 29: Among these, contrails most likely have the largest impact on the TOA radiation budget (Burkhardt and Kärcher, 2011; Brasseur et al., 2016).**

**Response:**

*We have incorporated this change.*

- **Line 37: contrails and contrail cirrus**

**Response:**

*We have incorporated this change.*

- **Line 38: Please rephrase. It is more complicated than it should**

**Response:**

*We agree with the Referee's suggestion. We have revised this sentence as follows:*

**Line 44 – 46:**

**"The impact of persistent contrails and contrail cirrus on the TOA radiation budget is often quantified using the radiative forcing (RF) (Chen et al., 2000) or the effective RF (ERF) metric. In our case, RF is defined as the radiative impact of a cloud, calculated as the difference in radiative fluxes at TOA between a cloudy and and cloud-free atmosphere."**

- **Line 42: ERF is introduced but not explained or referenced**

**Response:**

*The Referee is right. We have revised the text as follows:*

**Line 47 - 48:**

**'ERF, in contrast, includes all tropospheric and land surface adjustments, whereas RF only includes the adjustment due to stratospheric temperature change (Smith et al., 2020).'**

- **Lines44-45: Listing the two models is not necessary. I would suggest removing them or otherwise adding citations.**

**Response:**

*The Referee is right. We have revised the text as follows:*

**Line 60:**

**'On a global scale, either general circulation models of the atmosphere, reanalyses data ...'**

– **Line 61: at the TOA**

**Response:**

*We have incorporated this change.*

– **Lines 42-67: Can be shortened. In some cases, too many details are given about individual publications.**

**Response:**

*We have shortened the paragraph originally spanning between Lines 50 to 60 as follows:*

**Line 65-68:**

**"On smaller spatial and temporal scales, studies have been carried out using polar-orbiting satellite observations, geostationary ones or a combination of both (Haywood et al., 2009; Wang et al., 2024; Dekoutsidis et al., 2023; Duda et al., 2004; Mannstein and Schumann, 2005; Graf et al., 2012; Schumann and Graf, 2013; Wang et al., 2023; Meijer et al., 2022). "**

– **Line 71: infrared**

**Response:**

*We have incorporated this change.*

– **Line 80: Figure 1 is a really nice graphic representation of the presented method. Consider referring to it in the text here.**

**Response:**

*We have revised the text as follows:*

**Line 93:**

**'A methodological flowchart of the approach is presented in Figure 1. '**

– **Line 97: Section 2 contains**

**Response:**

*We have incorporated this change.*

– **Lines 100-101: Section 5 validation?**

**Response:**

*The Referee is right. We have revised the text as follows:*

**Line 118 – 119:**

**"A detailed validation of the Rapid Contrail-RF Estimation Approach is provided in Section 5. Finally, conclusions and future perspectives are discussed in Section 6."**

– **Lines 103-105: Please rephrase.**

**Response:**

*We have rephrased the sentence and now it reads as:*

**Line 121-122:**

**"In this study, the Rapid Contrail-RF Estimation Approach is deployed to generate RF maps for high-altitude ice clouds above the geographic area of interest, following these three initial steps: "**

– **Lines 110-117: Add references**

**Response:**

*We have added the following additional references:*

*- Schmetz, J., Pili, P., Tjemkes, S., Just, D., Kerkmann, J., Rota, S., and Ratier, A.: An introduction to Meteosat second generation (MSG), Bulletin of the American Meteorological Society, 83, 977–992, 2002.*

*- Roberto, N.: Satellite analysis of cloud characteristics at different temporal and spatial scales using visible and infrared wavelengths., 2010.*

- *Huckle, R. and Fischer, R. P. D. H.: Determination of clouds in MSG data for the validation of clouds in a regional climate model, Ph.D. thesis, Verlag nicht ermittelbar, 2009*

– **Line 115: which become larger**

**Response:**

*We have incorporated this change.*

– **Lines 115-117: Since the HRV is not used there is no added value to refer to it here.**

**Response:**

*We have removed the information about HRV.*

– **Line120: Remove "initially"**

**Response:**

*We have removed the word "initially".*

– **Line 122: distinguished to other**

**Response:**

*We have incorporated this change.*

– **Line 123: also successfully utilized the Dust RGB**

**Response:**

*We have incorporated this change.*

– **Line 126: persistent contrails were**

**Response:**

*We have incorporated this change.*

– **Line 130: Consider including a Dust RGB image from one of the days as an example or refer to Fig7**

**Response:**

*We refer to Figure 7 as follows:*

**Line 137:**

**"… to other cloud types (see Figure 7)…"**

– **Line 170: Liquid water cloud**

**Response:**

*We have incorporated this change.*

– **Line 173: accuracy of such a simplification**

**Response:**

*We have incorporated this change.*

– **Line 182: liquid water cloud**

**Response:**

*We have incorporated this change.*

– **Line 186 and everywhere relevant: liquid water cloud**

**Response:**

*We have incorporated this change in the whole manuscript including the captions of Figures 11, 15, and 16 and Table 1.*

– **Line 189: cite zenodo**

**Response:**

*We have added the Zenodo link as a hyperlink to the word Zenodo in the manuscript.*

   – **Line 196: Equation preferably centered**

**Response:**

*We choose to leave the Equation as it is.*

   – **Line 203: while keeping the following constant**

**Response:**

*We have incorporated this change.*

   – **Line 219: Reference**

**Response:**

*We have added the Zenodo link as a hyperlink to the word Zenodo in the manuscript.*

   – **Lines 224 & 229 & 235 and elsewhere: Figures and Tables are not capable of showing. In Figures the authors present, provide, showcase etc.**

**Response:**

*We have incorporated these changes.*

   – **Line 226: at the TOA due to cloud reflectivity**

**Response:**

*We have incorporated this change.*

   – **Line 226: Thicker or optically thicker?**

**Response:**

*We have changed the sentence as:*
**"...becomes optically thicker…"**

   – **Lines 227,231,232 and elsewhere: add optical thickness or optically thick**

**Response:**

*We have incorporated these changes.*

   – **Line234: does not have as large an effect as**

**Response:**

*We have incorporated this change.*

   – **Figure 2: Y axis could stop at -700 for better visualization and CER legend can be moved to below the figure since it applies to all plots (same Figure 3)**

**Response:**

*Figure 2 demonstrates Rfsol for an ice cloud above a water cloud with a cloud optical thickness of 5. However, this was not consistent with our intention to illustrate the behavior of Rfsol over the ocean. We have updated Figure 2 and applied the recommended modification to the CER legend in both figures.*

   – **Chapter 2: Could be shortened. In some cases too many details are provided that could be substituted by references.**

**Response:**

*We have shortened all the subsections of Chapter 2, expect for Section 2.3, where we detail the RT simulation and consequently the construction of LUTs.*

   – **Line 280: Not needed**

**Response:**

*We have deleted this sentence.*

    – **Line 284: Parenthesis missing**

**Response:**

*We have added the correction.*

    – **Line 299: Distinguishing between natural and**

**Response:**

*We have incorporated this change.*

    – **Line 301: Same as above**

**Response:**

*We have re-named the Section 3.2 as "Distinguishing between ice and cirrus clouds" as well as adding the following sentence:*

**Line 302:** *"To emulate the detection of persistent contrails, among naturally occurring and contrail clouds, ..."*

    – **Line 305: On average**

**Response:**

*We have incorporated this change.*

    – **Line 313: Section 5 does not need to be introduced here**

**Response:**

*We agree with the Referee and we have deleted this sentence.*

    – **Line 339: Is most commonly expected to result in cooling**

**Response:**

*We have incorporated this change.*

    – **Line 366: (Subsection 5.0.3)**

**Response:**

*We have added (Subsection 5.3) in the sentence.*

    – **Line 382: A metric supporting the "good agreement" statement would be useful**

**Response:**

*The metric is the correlation coefficient and slope, which are close to unity. We have added actual value of these metrics to support our statement:*

**" As it can be seen, for all the scene scenarios in the SW wavelength range, overall good agreement is found with the correlation coefficient (R) ranging from 0.97 to 1.00 and slope (s) from 0.93 to 0.97, with the exception of a few comparison points. "**

    – **Line 385: As a percentage the errors could be better understandable**

**Response:**

*We agree with the Referee, so we state the RMS error percentage.*

    – **References: Be sure to conform to the AMT guidelines. I believe a doi is required for every article.**

**Response:**

*We have updated the full reference list, including DOIs and ensured that journal names are abbreviated according to the Journal Title Abbreviations by Caltech Library.*

---

## Referee Report (RR1)

Manuscript number: egusphere-2025-697
Full title: Satellite-based estimation of contrail cirrus cloud radiative forcing derived through a Rapid Contrail-RF Estimation Approach
Author(s): Dimitropoulou E. et al.

The authors have improved the manuscript substantially, and it looks better now. I understand that the current manuscript is to show the performance of the analysis system for the contrail-induced radiative forcing estimation (i.e., the Rapid Contrail-RF Estimation Approach), and the authors specified that an independent validation of the contrail-detection method is beyond the scope of the present work and is future work. However, there are still a significant number of descriptions stating that the authors are investigating the radiative forcing of contrails (e.g., L102, L131, L303, and many more). The authors should first define what the detected cloud cases are through the CTP filter (perhaps "high-altitude ice clouds" or "potential contrails"), and they should consistently use this terminology in the following sections. The topic in the present paper is suitable for *Atmospheric Measurement Techniques (AMT)*. As long as the above-mentioned inconsistency is resolved, it can be published. Please find the minor comments below for potential improvement of the manuscript.

**Minor comments**
1. Title: "Satellite-based estimation of contrail cirrus cloud radiative forcing derived through a Rapid Contrail-RF Estimation Approach." The red-highlighted part is no longer relevant to the work. My impression is that the manuscript is focused on the validation of the Rapid Contrail-RF Estimation Approach. Please revise the title to be more relevant to the work.
2. Abstract, Page 1, L12 "seasons": The authors only analyze 6 days of data, which is insufficient to resolve the seasonal variability of the cloud variables due to substantial daily variation. Please delete this word.
3. Page 3, Lines 61 "between": Are there only two relevant studies? If there are more than two studies, suggest the authors rephrase it with "among".
4. Page 3, Line 76 "In (Driver et al., 2025)": This is a format error. Please correct it.
5. Page 13, Line 297 "Distinguishing between ice and cirrus clouds": Cirrus clouds are part of ice clouds. Please rephrase it to make it unambiguous.

---

## Referee Report (RR2)

Review of: **"Satellite-based estimation of contrail cirrus cloud radiative forcing derived through a Rapid Contrail-RF Estimation Approach"** by Dimitropoulou et al.

The authors have provided in-depth, scientific and satisfactory replies to all of my questions and remarks after the first round of reveiews and have changed and adapted their manuskript according to the suggestions and technical corrections.

My evaluation of the scientific significance of the manuscript, its relevance to the journal as well as its added value remain unchanged from the first round of reviews. Considering all the above in my opinion the manuscript can be acepted for pulbication as is.

---

## Author Response (AR2)

**Response to Anonymous Referee #3 for the second round of reviewing of 'Satellite-based estimation of contrail cirrus cloud radiative forcing derived through a Rapid Contrail-RF Estimation Approach' by Dimitropoulou et al. (https://doi.org/10.5194/egusphere-2025-697)**

**Referee:**
The authors have improved the manuscript substantially, and it looks better now. I understand that the current manuscript is to show the performance of the analysis system for the contrail-induced radiative forcing estimation (i.e., the Rapid Contrail-RF Estimation Approach), and the authors specified that an independent validation of the contrail-detection method is beyond the scope of the present work and is future work. However, there are still a significant number of descriptions stating that the authors are investigating the radiative forcing of contrails (e.g., L102, L131, L303, and many more). The authors should first define what the detected cloud cases are through the CTP filter (perhaps "high-altitude ice clouds" or "potential contrails"), and they should consistently use this terminology in the following sections. The topic in the present paper is suitable for Atmospheric Measurement Techniques (AMT). As long as the above-mentioned inconsistency is resolved, it can be published. Please find the minor comments below for potential improvement of the manuscript.

**Response:**
We would like to thank Referee #3 for the valuable comments provided after reviewing the revised manuscript. We fully agree that referring to the selected clouds as "contrails" can be misleading and erroneous. To ensure consistency, we have revised the terminology throughout the manuscript. In all relevant sections, the previous references to "contrails" have been replaced with "high-altitude ice clouds" or "potential contrails", depending on the context. All corresponding changes are indicated in the marked-up PDF.

**Minor comments**

**Referee:**
1. Title: "Satellite-based estimation of contrail cirrus cloud radiative forcing derived through a Rapid Contrail-RF Estimation Approach." The red-highlighted part is no longer relevant to the work. My impression is that the manuscript is focused on the validation of the Rapid Contrail- RF Estimation Approach. Please revise the title to be more relevant to the work.

**Response:**
We agree with the Referee and that the original title no longer accurately reflects the content of the revised manuscript. We have updated the title of our manuscript as follows:
"**Satellite-based estimation of high-altitude ice cloud radiative forcing** derived through a Rapid Contrail-RF Estimation Approach."

**Referee:**
2. Abstract, Page 1, L12 "seasons": The authors only analyze 6 days of data, which is insufficient to resolve the seasonal variability of the cloud variables due to substantial daily variation. Please delete this word.

**Response:**
We agree that the use of the word "seasons" is not justified by the presented results. We have therefore removed this word.

**Referee:**
3. Page 3, Lines 61 "between": Are there only two relevant studies? If there are more than two studies, suggest the authors rephrase it with "among".

**Response:**
More than two relevant studies are referenced in this sentence. We have replaced the word "between" with "among" to improve accuracy.

**Referee:**

4. Page 3, Line 76 "In (Driver et al., 2025)": This is a format error. Please correct it.

**Response:**

We have corrected the citation format error accordingly.

**Referee:**

5. Page 13, Line 297 "Distinguishing between ice and cirrus clouds": Cirrus clouds are part of ice clouds. Please rephrase it to make it unambiguous.

**Response:**

We agree with the Referee that a rephrasing of the section title is essential, since cirrus clouds are a subset of ice clouds. To improve clarity, we have updated the section title to "Selection of high-altitude ice clouds".